# Stem cell homeostasis regulated by hierarchy and neutral competition

Asahi Nakamuta[1,2,6], Kana Yoshido[1,6] & Honda Naoki [1,3,4,5✉]

Tissue stem cells maintain themselves through self-renewal while constantly supplying differentiating cells. Two distinct models have been proposed as mechanisms of stem cell homeostasis. According to the classical model, there is hierarchy among stem cells, and master stem cells produce stem cells by asymmetric division; whereas, according to the recent model, stem cells are equipotent and neutrally compete. However, the mechanism remains controversial in several tissues and species. Here, we developed a mathematical model linking the two models, named the hierarchical neutral competition (hNC) model. Our theoretical analysis showed that the combination of the hierarchy and neutral competition exhibited bursts in clonal expansion, which was consistent with experimental data of rhesus macaque hematopoiesis. Furthermore, the scaling law in clone size distribution, considered a unique characteristic of the recent model, was satisfied even in the hNC model. Based on the findings above, we proposed the criterion for distinguishing the three models based on experiments.

[1] Laboratory of Theoretical Biology, Graduate School of Biostudies, Kyoto University, Yoshidakonoecho, Sakyo, Kyoto 606-8315, Japan. [2] Faculty of Science, Kyoto University, Yoshidakonoecho, Sakyo, Kyoto 606-8315, Japan. [3] Laboratory of Data-driven Biology, Graduate School of Integrated Sciences for Life, Hiroshima University, Kagamiyama, Higashi-hiroshima, Hiroshima 739-8526, Japan. [4] Kansei-Brain Informatics Group, Center for Brain, Mind and Kansei Sciences Research (BMK Center), Hiroshima University, Kasumi, Minami-ku, Hiroshima 734-8551, Japan. [5] Theoretical Biology Research Group, Exploratory Research Center on Life and Living Systems (ExCELLS), National Institutes of Natural Sciences, Okazaki, Aichi 444-8787, Japan. [6]These authors contributed equally: Asahi Nakamuta, Kana Yoshido. ✉email: nhonda@hiroshima-u.ac.jp

All cells in our body are derived from stem cells, which maintain their populations through self-renewal while constantly producing differentiated cells to achieve homeostasis in any tissue. This property of tissue stem cells is vital for homeostasis in organisms, and defects in stem cell homeostasis cause various diseases such as cancer and infertility[1–3]. In stem cell homeostasis, fate asymmetry is achieved, in which a half of the daughter cells from the entire stem cell population is retained as stem cells and the other half is directed toward differentiation. To date, two distinct models have been suggested to explain the mechanisms of stem cell homeostasis, i.e., how the cell fate asymmetry is achieved[4,5]: the hierarchical model and the neutral competition (NC) model.

However, the mechanisms of stem cell homeostasis in many species and tissues remain controversial.

The hierarchical model has traditionally been accepted. In the model, there are two hierarchies of stem cells, namely master and non-master stem cells (Fig. 1a). Master stem cells are the most undifferentiated stem cell population consisting of a limited number of stem cells, while non-master stem cells are more differentiated and irreversibly directed toward differentiation. For stem cell homeostasis, one master stem cell generates one master stem cell and one non-master stem cell through asymmetric division (Fig. 1b). Such asymmetry at the single-cell level achieves fate asymmetry in the entire stem cell population, where loss of stem cells by their differentiation is compensated by asymmetric division of master stem cells, ensuring the maintenance of the

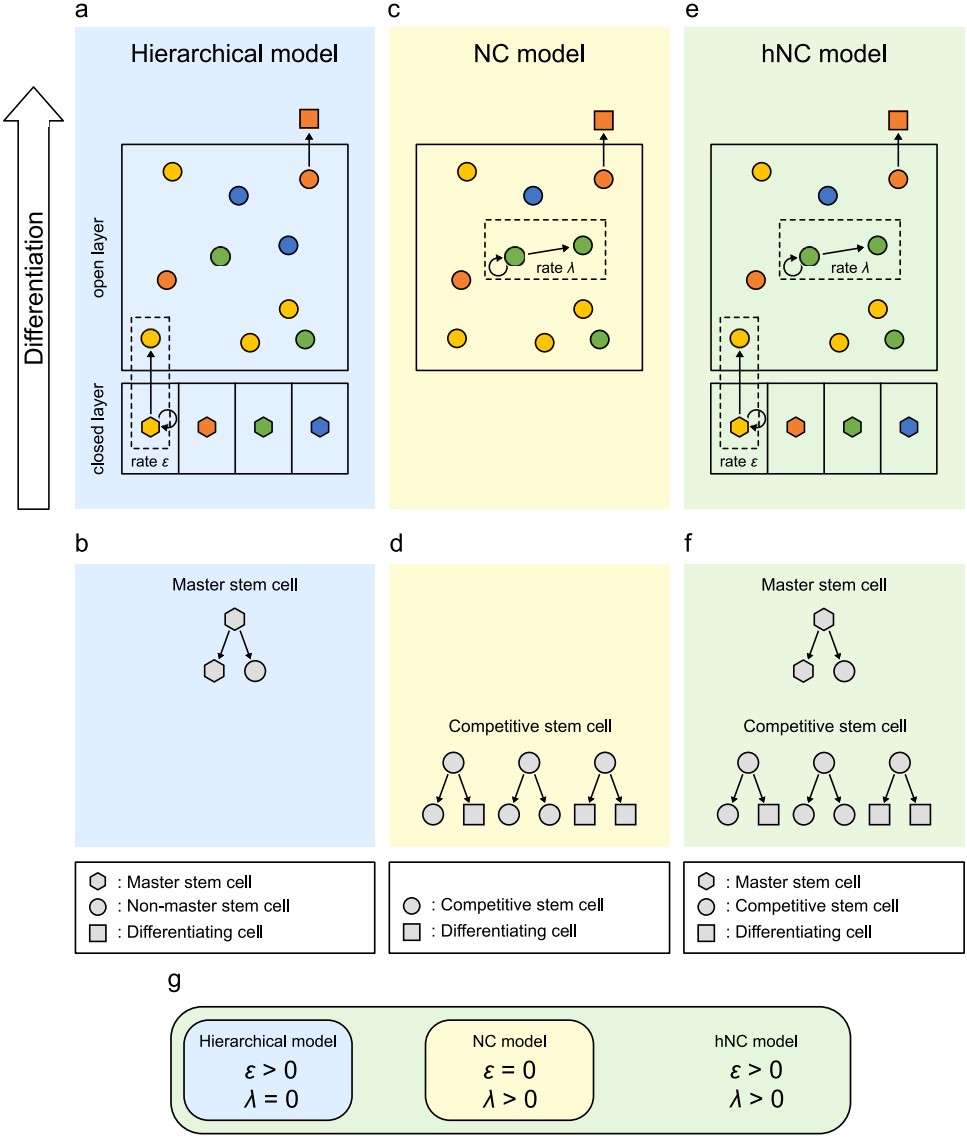

**Fig. 1 Three distinct biological models for stem cell homeostasis. a** Hierarchical model. Loss of stem cells is compensated for by asymmetric division of master stem cells (hexagon) in the closed layer. In the open layer, non-master stem cells (circles) do not compete with each other and are directed to differentiation. Each color indicates each clone derived from the common master stem cell. **b** Diagram of patterns of stem cell divisions in the hierarchical model. **c** NC model. Loss of stem cells is compensated for by proliferation of competitive stem cells (circles) in the open layer. Each color indicates each clone derived from the common competitive stem cell. **d** Diagram of patterns of stem cell divisions in the NC model. **e** Hierarchical neutral competition (hNC) model. Loss of stem cells is compensated for by the asymmetric division of master stem cells (hexagon) in the closed layer and proliferation of competitive stem cells (circles) in the open layer. Each color indicates each clone derived from the common master stem cell. **f** Diagram of patterns of stem cell divisions in the hNC model. **g** Relationships between the three models. The mathematical model seamlessly represents all three biological models based on two parameters, $\varepsilon$ and $\lambda$, which represent the proliferation rates of master stem cells and competitive stem cells, respectively.

stem cell population. In *Drosophila*, stem cells in the germline and developing central nervous system have been demonstrated to undergo invariant asymmetric cell divisions, giving rise to one stem cell and one differentiating cell[4,5], strongly supporting the hierarchical model. The model has also been considered extensively in mammalian stem cell homeostasis. In classical models of epidermal and spermatogenic stem cells, a limited number of stem cells are thought to participate in the maintenance of homeostasis[6–8].

Another distinct model, known as the neutral drift model or stochastic model, has recently attracted the attention of researchers[9]. We refer to this model as the NC model to focus on the clonal expansion process occurring through neutral cell-cell competition. In the NC model, all stem cells are equipotent, without the hierarchy among stem cells assumed in the hierarchical model, and we call all these stem cells "competitive stem cells" (Fig. 1c). The model assumes that the fate of each stem cell is not fixed: a stem cell generates two stem cells or two differentiating cells by symmetric division or one stem cell and one differentiating cell by asymmetric division (Fig. 1d). Therefore, fate asymmetry is achieved at the cell population level rather than at the single-cell level, and loss of one stem cell is compensated for by the symmetric division of the other stem cell into two stem cells. Thus, stem cells neutrally compete with each other, leading to neutral drift in clonal expansion. The NC model was originally suggested based on the observation of the random fate of stem cells[10,11] and several inducible genetic labeling studies[12,13]. Recently, Klein and Simons performed a mathematical analysis of the NC model and demonstrated that stem cell clonal expansion adheres the scaling law, in which the scaled probability distribution of the stem cell population size of each clone is universal over time[14]. Several experimental studies involving long-term lineage tracing of stem cell clones have confirmed the scaling law in mouse epidermal, intestinal, and spermatogenic tissues, supporting the NC model as the mechanism of stem cell homeostasis[15–18]. Therefore, the NC model has recently been considered another potential mechanism of stem cell homeostasis in mammalian tissue.

However, a phenomenon that cannot be predicted by either existing model alone has been reported. A study using genetic barcoding of spermatogonial stem cells in mice reported that offspring were derived non-randomly from all clonal types of spermatogonial stem cells, and that offspring of specific clones were periodically observed, suggesting that the population size of each stem cell clone exhibited transient burst-like patterns repeatedly[19]. Similarly, in hematopoiesis in primates, burst-like dynamics was observed in repopulation kinetics of hematopoietic stem and progenitor cells (HSPCs) by clonal tracking of transplanted HSPCs[20]. In the hierarchical model, the clone population size is stably maintained because each master stem cell compensates for the loss of stem cells without competition, whereas in the NC model, a few clones dominate the population and other clones end up in extinction owing to the neutral evolution of the stem cell clones. As both the hierarchical model and the NC model cannot explain the burst-like dynamics of stem cell clones, there should be a missing factor for understanding stem cell homeostasis.

To address the controversy with regard to the mechanism of mammalian stem cell homeostasis, we focused on the fact that two existing models are not necessarily contradictory. In fact, both models have been supported by data from different experimental systems, leading to different results. Thus, the same phenomenon has been potentially observed from different aspects. Consequently, in the present study, we present a comprehensive mathematical model that seamlessly links the two existing models. In addition, the mathematical model represents an intermediate model that we named the hierarchical neutral competition (hNC) model, in which both the supply of stem cells by master stem cells and competition between stem cell clones are compatible. Through numerical simulation and mathematical analysis, we revealed that the hNC model exhibited burst-like patterns in the clone dynamics of stem cells. Furthermore, we showed that the scaling law in clone size distribution, which has been proposed as an indicator of the NC model, was satisfied not only in the NC model but also in the hNC model. Based on the findings above, we proposed the criterion for distinguishing the three models, experimentally.

## Results

**General model for stem cell homeostasis**. To examine the effect of the presence of master stem cells and/or neutral competition among stem cells on clonal dynamics in stem cell homeostasis, we developed a mathematical model that comprehensively accounts for the hierarchical, NC, and their intermediate model named as the hNC model (Fig. 1a–f). The hNC model has two hierarchies of stem cells, similar to those in the hierarchical model: master stem cells in the closed layer and competitive stem cells in the open layer, while it also has neutral competition among competitive stem cells, similar to the NC model (Fig. 1e). In the closed layer, master stem cells supply competitive stem cells to the open layer while undergoing self-renewal via invariant asymmetric divisions. In the open layer, competitive stem cells compete while producing differentiating cells; when a competitive stem cell is lost through differentiation or apoptosis, it is compensated for either through supply from master stem cells or symmetric division of other competitive stem cells at rates of $\varepsilon$ and $\lambda$, respectively (Fig. 1f). It is widely accepted that more differentiated stem cells are more actively cycling with high proliferation, such that we considered the proliferation rate of the master stem cell to be lower than that of the competitive stem cell, i.e., $\varepsilon \leq \lambda$.

In the simulation of our model, at each step, one competitive stem cell was randomly selected for differentiation and excluded from the open layer. Subsequently, a master stem cell or competitive stem cell was selected with a weighted probability defined by the parameters $\varepsilon$ and $\lambda$, which supplies one competitive stem cell to maintain the total number of stem cells. From a theoretical perspective, this model was extended from the Moran process[21], which generally describes the evolutionary population dynamics of two species, by introducing multiple species and an external supply from master populations outside of competition (see Methods).

The mathematical model can represent three biological models by changing parameters $\varepsilon$ and $\lambda$ (Fig. 1g). The condition under which competitive stem cells do not proliferate ($\lambda = 0$), indicating the absence of neutral competition, corresponds to the hierarchical model. In contrast, the condition under which master stem cells do not supply competitive stem cells ($\varepsilon = 0$), indicating the absence of master stem cells, corresponds to the NC model. Additionally, the mathematical model represents the intermediate condition between the hierarchical and NC models, named the hNC model, that is $\lambda > 0$ and $\varepsilon > 0$.

**Clonal bursts were generated in the hNC model**. To investigate the dynamics of the three biological models (hierarchical, NC, and hNC models) for explaining stem cell homeostasis, we performed numerical simulations of our mathematical model using different values for the proliferation rate of master stem cells $\varepsilon$ and that of competitive stem cells $\lambda$. In the hierarchical model ($\varepsilon > 0$, $\lambda = 0$), no clone dominated the open layer, and the populations of all clones fluctuated around the averages because master stem cells continuously compensated for the loss of stem

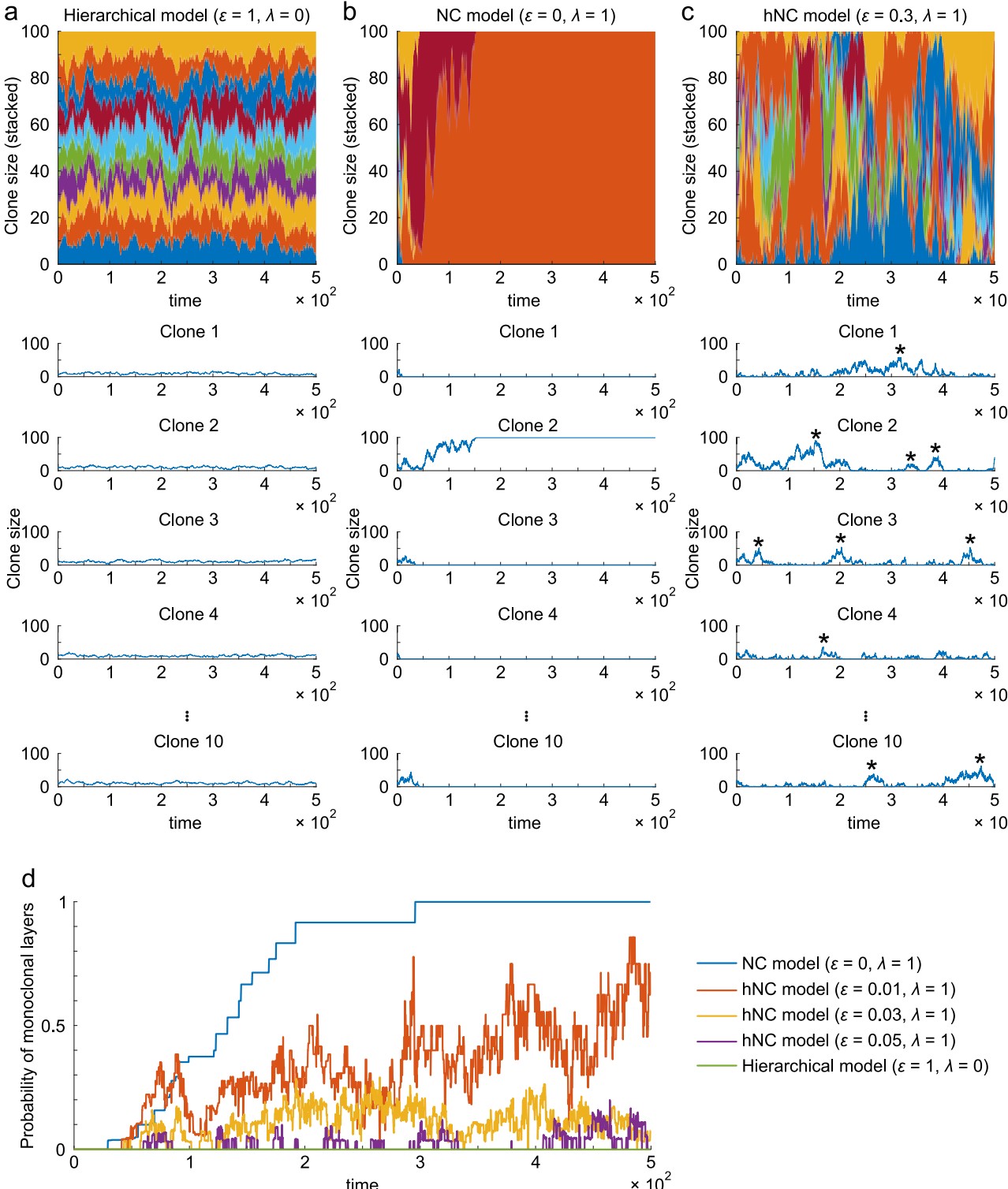

**Fig. 2 Clonal expansion in the hierarchical, NC, and hNC model. a–c** Time-series of clone sizes with different proliferation rates of master stem cells ($\varepsilon$) and competitive stem cells ($\lambda$). In the simulation, there were 10 types of clones in 100 competitive stem cells in the open layer, in which the clone size of every clone was initially uniform, i.e., $n_k = 10$ at $t = 0$. Asterisks indicate representative bursts. In the top panels, the fraction of each clone is displayed by a different color. In the lower panels, the results of five representative clones within 10 clones are displayed. **d** The probability of monoclonal conversion in open layers over time. We repeated simulations 1000 times, and the probability that the open layer became monoclonal at each timepoint is displayed.

cells from the closed layer (Fig. 2a). In contrast, in the NC model ($\varepsilon = 0$, $\lambda > 0$), only one clone dominated the open layer, and all other clones were extinguished through neutral, random drift of clonal populations (Fig. 2b). Interestingly, in the hNC model ($0 < \varepsilon \leq \lambda$), each clone population showed intermittent and

repeated expansion and contraction, like the bursts (Fig. 2c). These clonal bursts were generated by the mechanism that each clone was headed to domination or extinction by neutral drift; however, the two extreme results were not eventually achieved

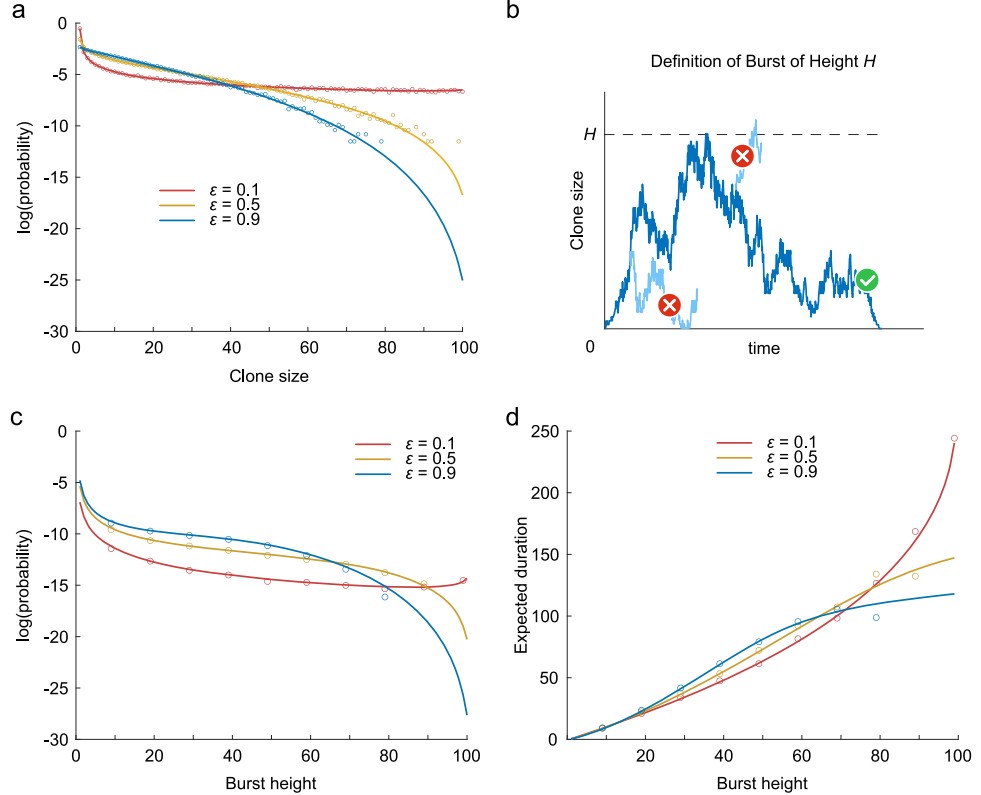

**Fig. 3 Clone size distribution and clonal bursts depending on the proliferation rate of master stem cells in the hNC model. a** Stationary probability distribution of clone size with different proliferation rates of master stem cells $\varepsilon$. The solid line and dots indicate the probability distributions derived through mathematical analysis (equation [7]) and numerical simulation, respectively. **b** Definition of clonal bursts. A burst of height $H$ is defined as the dynamics in which the clone size changes from 0 to $H$ without returning to 0, and then returns to 0 without reaching $H + 1$. **c** Probability of burst generation of each height $H$ starting at clone size 0 depending on the proliferation rates of master stem cells $\varepsilon$. Solid lines indicate the analytical solution for the probability (equation [15]). Dots indicate the probability calculated by numerical simulation. **d** Expected duration of a clonal burst of each height $H$ starting at clone size 0 depending on the proliferation rates of master stem cells $\varepsilon$. Solid lines indicate the analytical solution of the expected duration (equation [22]). Dots indicate the average duration of a clonal burst of each height calculated by the numerical simulation.

because of the infrequent supply from master stem cells, whose proliferation rate was lower than that of competitive stem cells.

As another characteristic of clonal expansion, we examined monoclonal conversion, a phenomenon in which only one clone dominates the stem cell population. Using a simulation, we quantified the probability of monoclonal conversion in the three models (Fig. 2d). Monoclonal conversion did not occur in the hierarchical model, whereas it occurred in the NC model; that is, the probability of monoclonal conversion increased over time and eventually converged to one in the NC model. In contrast, in the hNC model, the probability of monoclonal conversion increased over time and eventually reached and fluctuated around certain constants, approaching a larger value with a lower proliferation rate of master stem cells $\varepsilon$.

**Clonal bursts were caused by low proliferation of master stem cells**. We mathematically analyzed how clonal bursts were generated, as shown in Fig. 2, based on the theory of stochastic processes. We first derived the stationary probability distribution of the clone size (see Methods). We found that a lower proliferation rate of master stem cells $\varepsilon$ was associated with a higher probability of a large clone size (Fig. 3a), implying the existence of higher clonal bursts. Next, we directly analyzed the properties of clonal bursts. We defined a clonal burst (Fig. 3b) and derived the probability that a clonal burst of a certain height is generated and the expected duration of a clonal burst at a certain height (see Methods). By this theoretical analysis, we found that a lower

proliferation rate of master stem cells increased and decreased the probabilities of larger and smaller bursts, respectively (Fig. 3c and simulation results in Supplementary Fig. 1). In addition, we observed that a lower proliferation rate of master stem cells increased and decreased the durations of larger and smaller bursts, respectively (Fig. 3d and simulation results in Supplementary Fig. 1). These mathematical analyses were validated by confirming that the analytical solutions were reproduced in numerical simulations (Fig. 3a, c, d). Taken together, the proliferation rate of master stem cells relative to that of competitive stem cells dominantly influences the height and duration of stem cell clonal bursts.

**Scaling law was satisfied in both the NC and hNC models**. The existence of clonal bursts should be an indicator of the hNC model, consisting of hierarchy and neutral competition among stem cells, because the existing hierarchical and NC models cannot generate a clonal burst. On the other hand, Klein and Simons theoretically suggested that the NC model can be distinguished from the hierarchical model using the scaling law of the clone size distribution, in which the probability distribution of the stem cell population size of each clone, $P_n(t)$, time-independently obeys the universal scaled distribution[14]:

$$P_n(t) = \frac{1}{\langle n(t) \rangle} F\left( \frac{n(t)}{\langle n(t) \rangle} \right), \tag{1}$$

where $n(t)$ and $\langle n(t) \rangle$ denote the clone size and its average at time $t$, respectively, and $F(x)$ indicates the time-independent universal

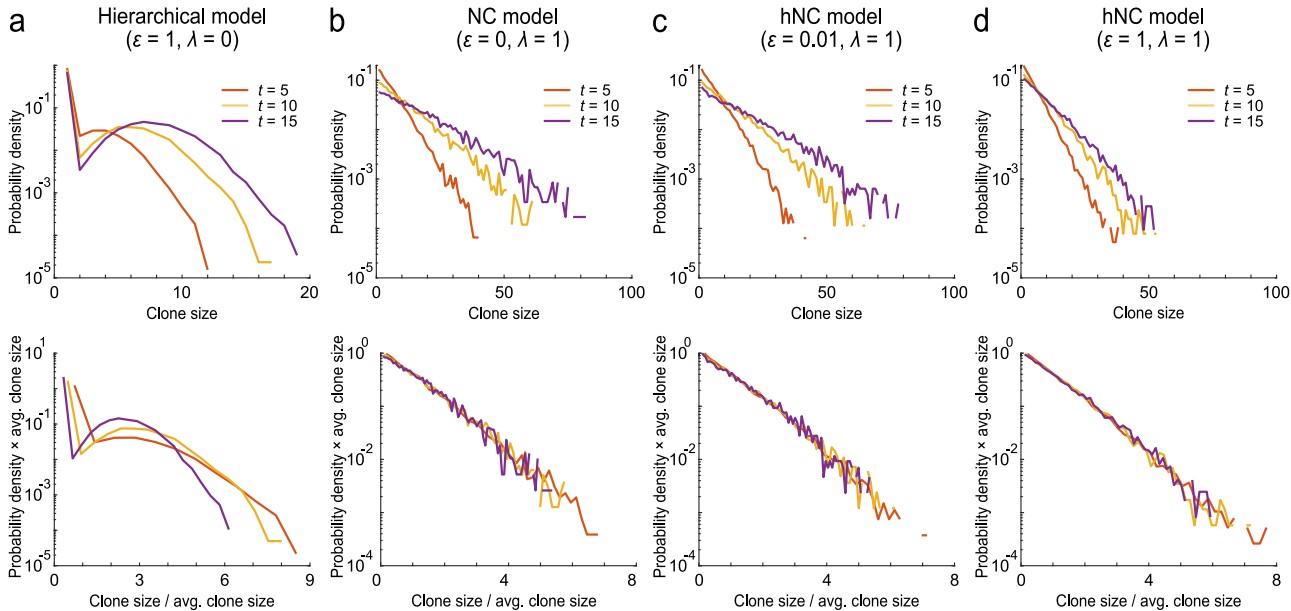

**Fig. 4 Scaling law of clone size distribution in the NC and hNC models.** Probability distribution of clone size calculated by the simulation mimicking the pulse-labeling experiment. Simulations were performed 100,000 times using 10 master stem cells and 100 competitive stem cells. Cells were randomly labeled from among the master and competitive stem cells. The clone size distributions of a labeled clone were plotted at different time points in the (**a**) hierarchical model, (**b**) NC model, and (**c**, **d**) hNC model with the two conditions. The upper and lower panels show the distributions before and after scaling by the average clone size at each time point, respectively, as equation (1).

function, which was determined from the spatial dimension of the tissue of interest. In our situation in the NC model, $F(x)$ was expected to obey an exponential distribution[18,22,23]. In fact, the dynamics of clonal expansion were monitored experimentally using lineage tracing of stem cells, which showed that clone size distribution followed the scaling law in several tissues in mice[15,16,18]. Notably, the scaling law holds only in the early phase of clonal expansion and not in the late phase, during which all stem cell populations converge to almost monoclonal.

To investigate whether clone size distribution in the hNC model satisfies the scaling law, we simulated our mathematical model with random labeling of one of the master and competitive stem cells. In the hierarchical model, the clone size distribution did not follow the scaling law (Fig. 4a). In contrast, the NC model obeyed a time-independent exponential distribution, which followed the scaling law (Fig. 4b). These observations are consistent with previous findings. Furthermore, we observed the scaling law of clone size distribution in the hNC model when the proliferation rate of the master stem cells was lower than or equal to that of the competitive stem cells (Fig. 4c, d), including the condition in which clonal bursts were generated, as shown in Fig. 2c and Supplementary Fig. 1. Similarly, clone size distribution did not follow the scaling law in the hierarchical model, but did in the NC and hNC models, when one of the master stem cells or one of the competitive stem cells was randomly labeled (Supplementary Figs. 2, 3). Furthermore, when the proliferation rate of master stem cells was much higher than that of competitive stem cells, which is probably not consistent with physiological conditions, the scaling behavior collapsed (Supplementary Fig. 4).

Overall, the results indicate that the scaling law of the clone size distribution can rule out the hierarchical model and be used as an indicator of neutral competition in clonal expansion, as in the NC and hNC models. However, the scaling law does not distinguish between the NC and hNC models; therefore, it is not possible to reject the presence of master stem cells based on the scaling law of the clone size distribution.

**Experimental data suggesting the hNC model in primate hematopoiesis.** To investigate the feasibility of the hNC model in stem cell homeostasis using experimental data, we focused on primate hematopoiesis[20]. A previous study performed clonal tracking analysis of hematopoietic stem and progenitor cells (HSPCs) following transplantation of vector-marked HSPCs, by which clones were distinguished and their populations were quantified at several time points.

Using their data, we generated a heatmap showing a time-series of clone sizes, where clone indexes were sorted by clustering of their temporal patterns (Fig. 5a). When focusing on each clone corresponding to each row, the population sizes of most clones increased and decreased repeatedly, suggesting burst-like dynamics, as observed in the hNC model. Furthermore, the hNC model reproduced the experimental results (Fig. 5b). In the simulation, we used a realistic parameter set estimated from the experimental results (see Methods for details) and sampled clone sizes at several timepoints corresponding to experimental intervals. In addition, we examined the clone size distribution in their experimental data (Fig. 5c). We then observed that the distributions at different timepoints had almost similar profiles, suggesting that clonal expansion dynamics had converged to stationary distribution in all timepoints. We also confirmed that both the analytical solution (as shown in Fig. 3a) and the simulation of the hNC model with the same realistic parameters generated similar profiles of the stationary clone size distribution (Fig. 5d). Overall, the primate hematopoietic stem cells were potentially maintained based on the hNC model.

**Experimentally distinguishing the three models.** Finally, we proposed a criterion for determining the mechanism of stem cell homeostasis in each tissue, that is, a criterion for distinguishing the hierarchical, NC, and hNC models using experimental data (Fig. 6a). First, to distinguish the hierarchical model from the other two models, one needs to examine whether the scaling law of clone size distribution holds using lineage tracing techniques, because we showed that the scaling law was an indicator of

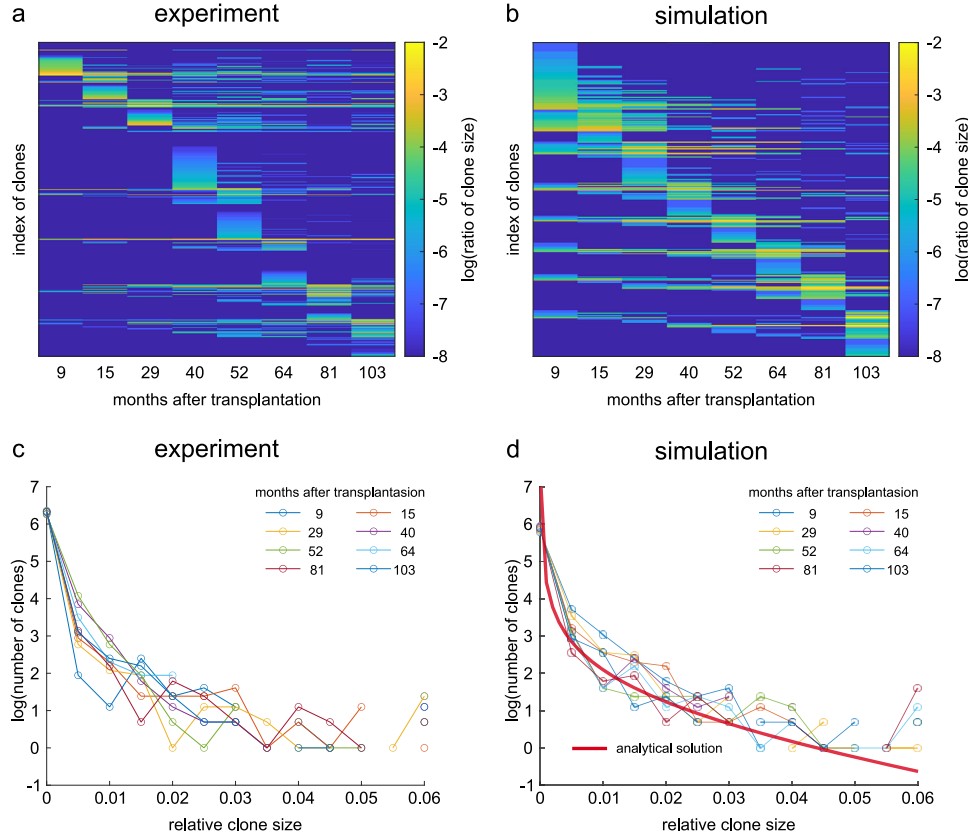

**Fig. 5 Comparison of clonal expansions between experiments and simulations of the hNC model in primate hematopoiesis. a** Decade-long time-series of clone sizes of stem cells in primate hematopoiesis. The data are publicly available from Kim et al., *Cell Stem Cell* (2014)[20]. Clones were sorted by time points of maximum population size. **b** Time-series of clone sizes in the simulation of the hNC model. Parameters were $K = 420$, $N = 1000$, and $\varepsilon = 0.05$. $\varepsilon$ was estimated from the data in (**a**) (see Methods). One month corresponds to 1000 steps in the simulation. **c, d** Histogram of clone size at different timepoints in the experiment (**c**) and simulation (**d**). Red line in (**d**) indicates the analytical solution of stationary distribution of clone size.

neutral competition, as included in the NC and hNC models (Fig. 4). Next, to distinguish between the NC and hNC models, one needs to examine whether clonal bursts are observed during lineage tracing over the long term, as we showed that the clonal bursts are unique to the hNC model (Fig. 2).

This criterion can distinguish the three models but does not provide more detailed information on the hNC model. As shown in Figs. 2 and 3, the proliferation rate of master stem cells, $\varepsilon$, governs burst-like dynamics, and thus, considerably influences stem cell homeostasis in each tissue type. Therefore, it is important to quantify $\varepsilon$ in the hNC model. Here, we propose two methods for estimating the proliferation rate of master stem cells, $\varepsilon$, in the hNC model, from experimental data. The first method was based on the Shannon index $H$, which is used in ecology to measure species diversity[24]:

$$H = -\sum_{m=1}^{M} P_m \ln P_m, \qquad (2)$$

where $m$ and $M$ indicate the index of labeled clones and number of introduced labels, respectively, and $P_m$ indicates the fraction of $m$-th labeled clonal stem cells. $P_m$ can be measured by lineage tracing of multiple stem cells. Note that it is not necessary to label all stem cell clones if the total number of competitive stem cells $N$ is determined in other experiments. By simulation of our mathematical model, we showed how $\varepsilon$ and $N$ influence the Shannon index (Fig. 6b). Based on the results, given $N$, $\varepsilon$ can be estimated using the Shannon index, which can be quantified experimentally using equation (2). The second method is based on the average duration of all bursts observed during lineage

tracing. The average duration of all bursts of all heights increased with an increase in $\varepsilon$ (Fig. 6c), indicating that $\varepsilon$ can be estimated experimentally if $N$ is known. To detect clonal bursts and observe their durations, frequent and long-term lineage tracing is necessary.

Finally, we demonstrated the estimation of $\varepsilon$ from the experimental data of primate hematopoiesis based on the Shannon index. We calculated the Shannon index $H$ from the experimental data of primate hematopoiesis (Fig. 5) ($H = 3.64$). For the estimation of $\varepsilon$ from $H$, we plotted a contour line at $H = 3.64$ in the heatmap (Red line in Fig. 6b). As $N$ should be more than the number of clone types observed in the experiments ($N > 420$), we estimated $\varepsilon \simeq 0.05$ (see Methods for detail).

## Discussion

How is stem cell homeostasis achieved in each tissue type? To date, it remains unclear whether there are hierarchies among stem cells and master stem cells provide stem cells by asymmetric division (the hierarchical model), or whether equipotent stem cells neutrally compete (the NC model), to maintain the total number of stem cells. We addressed this question by developing a mathematical model that links two existing models and represents an intermediate model, the hNC model. Through simulation and mathematical analysis, we found unique clonal dynamics in the hNC model but not in the two existing models; the clone size exhibited transient bursts repeatedly. We also showed that clonal bursts were caused by low proliferation of master stem cells, where domination or extinction by neutral drift was achieved incompletely. Although the dominant behavior of the

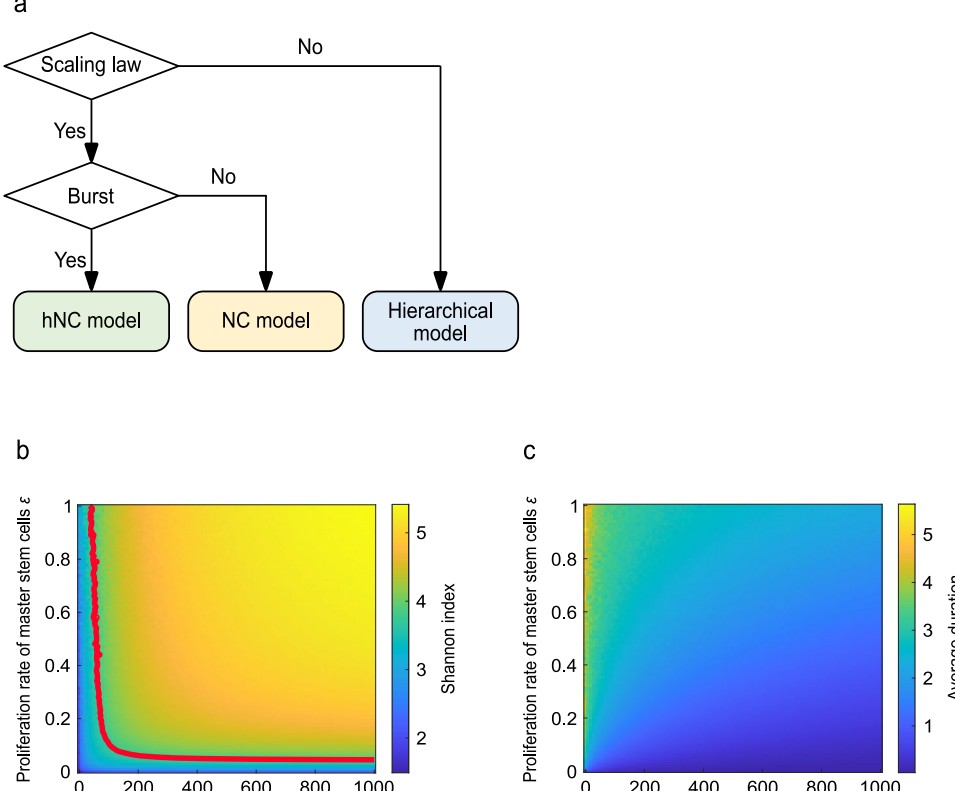

**Fig. 6 Criterion to experimentally distinguish the hierarchical, NC, and hNC models. a** Flow chart of criterion to distinguish the hierarchical, NC, and hNC models. Scaling law of clone size distribution can be used as an indicator to distinguish the hierarchical model and other two models. Clonal bursts can be used as an indicator to distinguish the NC model and hNC model. **b**, **c** Two types of experimentally-measurable variables depending on the proliferation rate of master stem cells, $\varepsilon$. The Shannon index $H$ representing clonal diversity (**b**) and average duration of all bursts (**c**) are shown. Given the total number of competitive stem cells $N$, $\varepsilon$ can be estimated using Shannon index and/or the average duration of all bursts, both of which can be measured through lineage tracing experiments of multiple clones. Red line in (**b**) indicates the contour at $H = 3.64$, which was calculated from the experimental data of primate hematopoiesis in Fig. 5a[20].

hNC model (neutral drift) was similar with the NC model, we considered the hNC model as the different biological model from the NC model, because the hNC model showed the distinct behavior in clonal expansion (clonal bursts). In addition, we also showed that clone size distribution in the hNC model followed the scaling law, proposed as an indicator of the NC model. In fact, the clonal bursts were found in experimental data of primate hematopoiesis[20], supporting the hNC model. Based on the distinct characteristics, we proposed a criterion for discriminating the three models using experimental data.

Hematopoietic stem cells have been largely classified into two hierarchical types: long-term and short-term hematopoietic stem cells, based on the stem cell properties[25]. They are both considered stem cells and have a hierarchical relationship. Furthermore, it was reported that adult hematopoiesis was largely sustained by short-term hematopoietic stem cells, rather than long-term hematopoietic stem cells[26]. This result suggests that in the hNC model, long-term and short-term hematopoietic stem cells correspond to master stem cells with a low proliferation rate and competitive stem cells with a high proliferation rate, respectively.

Some studies have reported findings that support the hNC model in hematopoiesis. Kim at al. examined the repopulation dynamics of transplanted HSPCs by clonal tracking of HSPCs in rhesus macaques for up to 14 years[20]. They demonstrated that different clones showed sequential burst-like expansions, peaking at different time points (Fig. 5a). A subsequent theoretical study

suggested that the burst-like dynamics are achieved by multi-step differentiation processes[27]. On the other hand, we demonstrated that the previous experimental data were explained by the hNC model; the burst-like dynamics and stationary clone size distribution were reproduced (Fig. 5). Note that burst-like dynamics observed in the present study were derived from large scale fluctuation of stem cell clonal populations based on the stochastic process, which is different from dynamic bursts achieved by the intrinsic properties of stem cells, such as the multi-step differentiation process. Therefore, hematopoietic stem cell homeostasis is regulated by the simple lineage structure with two stem cell populations (master and competitive stem cells, as in the hNC model), without the assumption of a complex structure, such as multi-step differentiation process.

There is also experimental evidence suggesting the hNC model in mammalian spermatogenesis. Kanatsu-Shinohara et al. examined the dynamics of mouse germline transmission following transplantation of spermatogonial stem cells with genetic labeling[19]. By analyzing genetic labels in offspring, they observed that offspring were derived non-randomly from all labeled clonal types of spermatogonial stem cells, and periodically observed offspring derived from specific clones. The results suggest that the population size of each stem cell clone increased and decreased repeatedly, exhibiting transient burst-like dynamics. In addition, the most undifferentiated stem cell population, that is, undifferentiated spermatogonia, was actively cycling without mitotic quiescence, whereas a more differentiated stem cell population of

differentiating spermatogonia partially underwent apoptosis during differentiation.

Considering the interpretation of the findings based on the hNC model, undifferentiated spermatogonia might correspond to master stem cells that are equally cycling. In contrast, differentiating spermatogonia might correspond to competitive stem cells, because they partially undergo apoptosis, implying the presence of cell competition. Therefore, in mouse spermatogenesis, master stem cells potentially supply competitive stem cells constantly and competitive stem cells compete with each other, which can lead to burst-like dynamics, as observed in the hNC model. Therefore, these experimental results support the hNC model, although the lineage of labeled stem cells should be observed directly to verify this model.

The mathematical model developed in the present study describes clonal dynamics, which consists of two types of stem cells: master and competitive stem cells. However, it has been recently reported that stem cells have heterogeneity among gene expression and properties as stem cells[25,28,29]. Therefore, stem cells may undergo multistage differentiation processes other than two compartments of master and competitive stem cells, as assumed in our mathematical model. Investigations of the additional processes could be insightful, considering that more complex lineage structures in stem cells can influence the clonal expansions in stem cell homeostasis[27]. However, as more complex models require more detailed assumptions related to lineage structure, which are still controversial in experimental studies, we assumed only two stem cell populations as a minimal and simple model of stem cell homeostasis similar to several previous theoretical studies[5,6,14].

It is also worth mentioning the biological interpretation of master and non-master stem cells in the hierarchical model. In this model where there is no competition among stem cells, we considered stem cells differentiated from master stem cells as non-master stem cells, and not as competitive stem cells (Fig. 1a). On the other hand, classically, the hierarchical model assumes that stem cells undergo invariant asymmetric divisions, which generate one stem cell and one transit-amplifying (TA) cell by each division. This has been observed in *Drosophila* spermatogenesis[5] and considered in the classical model of epidermal stem cell homeostasis[6]. However, considering more heterogeneous nature of stem cell populations than previously expected, it is more feasible that master stem cells supply other types of stem cells, even if stem cell homeostasis is regulated by the hierarchical model. Note that we did not introduce TA cells in the model, because we considered TA cells to correspond to cells out of the open layer; that is, TA cells are differentiated from non-master stem cells or competitive stem cells.

Stem cell homeostasis has been previously considered to be regulated by the hierarchical or NC model, and several studies have suggested that these two models can be distinguished based on the scaling law of clone size distribution[15–17]. However, it is theoretically possible that the hierarchy and neutral competition among stem cells are compatible, and we proposed the hNC model. Thus, it is not appropriate to identify the model of stem cell homeostasis only based on either of the evidence of hierarchy or that of neutral competition.

In recent studies, a generalized mathematical model of stem cell fate choice, which included the same condition as the hNC model, has been developed[30,31]. Their analysis showed that the lineage structure with hierarchy among stem cells also follows the scaling law in their clone size distribution depending on model parameters. The result indicates that scaling law of clone size distribution cannot rule out the presence of hierarchy among stem cells. The hNC model can be classified into the generalized hierarchical model based on their definitions. Indeed, consistent with their findings, we showed that the hNC model followed the scaling law in clone size distribution, because the NC and hNC models show almost similar behavior in the early phase after pulse-labeling because of the small population and low rate of proliferation of master stem cells. Therefore, multiple indicators should be used to identify the model of stem cell homeostasis, as shown in Fig. 6, and we could have overlooked the feasibility of the hNC model from data based on current experimental techniques.

When studying stem cell homeostasis, it is challenging to identify distinct characteristics of stem cells, which makes it difficult to define the cells in their natural context within tissues. In fact, it is difficult to determine which stem cell populations, such as master stem cells and competitive stem cells, have been actually observed by a widely-used gene marker for stem cells. In the present study, we proposed a criterion for distinguishing the three models by examining the presence or absence of scaling law of clone size distribution and bursts in stem cell clonal dynamics. Notably, the criterion does not depend on the strict definition of stem cells. To detect the scaling law and bursts, a minimum requirement is labeling the most undifferentiated stem cell population, which is master stem cells in the hNC and hierarchical models and competitive stem cells in the NC model. In other words, a gene marker that defines the cell population, including the most undifferentiated stem cells, is sufficient to distinguish the three models, which means we do not need to find a gene marker specific to the most undifferentiated stem cells. It is potentially easier to identify this population because some tissue stem cells, such as intestinal, epithelial, and spermatogenic stem cells, show distinct anatomical features. In such tissues, stem cells and differentiated cells change their physical positions in one direction during differentiation. This distinct characteristic may enable identification of stem cell populations, including the most undifferentiated cells.

## Methods

**Mathematical model of stem cell clone dynamics**. To examine stem cell clonal dynamics, we developed a mathematical model that comprehensively represents the hierarchical, NC, and hNC models. The mathematical model is based on the Moran process, which is a simple stochastic process describing population dynamics. The model is comprised of two types of stem cells: master stem cells in the closed layer and competitive stem cells in the open layer (Fig. 1a, c, e). The closed layer contains $K$ master stem cells, which are stably maintained and not subject to competition, and provide differentiated cells named as competitive stem cells to the open layer with a proliferation rate defined as $\varepsilon$. The open layer contains $N$ competitive stem cells, which are lost by their differentiation or apoptosis, and each loss is compensated by proliferation of other competitive stem cells in the open layer or master stem cells from the closed layer.

We simulated the population dynamics of distinct $K$ clones of competitive stem cells. Each elementary step of the simulation involved a loss event and compensation event. For loss, one of the $N$ competitive stem cells is selected at random. The selection probability of the $k$th clone is $p_k = n_k/N$, where $n_k$ indicates the population of the $k$-th clone in the open layer. For compensation, one stem cell is selected from the closed and open layers at random, based on proliferation rate of each clone; the selection probability of the $k$-th clone is $q_k = (\lambda n_k + \varepsilon)/(\lambda N + \varepsilon K)$. The total populations in the closed and open layers, that is, $K$ and $N$, are constant. As $N$ is constant, the total population of stem cells is consequently maintained. Note that this is different from the situation in Lotka-Volterra equations that are frequently used to model competitive dynamics in ecological communities and that the multivariate Moran process can be expanded to the model that corresponds to equilibrium Lotka-Volterra phenomenology[32].

To ensure time-scale consistency among the mathematical models with different parameters, the time scale was calibrated based on the average time taken per step as $t = m/(\lambda N + \varepsilon K)$, where $m$ indicates the number of steps in the simulation.

**Master equation of Moran process**. To analyze clonal expansion in the Moran process, we focused on one clone of $K$ clones in the open layer. The dynamics of a clone of interest can be described using the master equation (3) (Supplementary Fig. 5):

$$P_n^{m+1} = r_+(n-1)P_{n-1}^m + r_-(n+1)P_{n+1}^m + \left(1 - r_+(n) - r_-(n)\right)P_n^m, \quad (3)$$

where $P_n^m$ denotes the probability that the size of clones of interest is $n$ in step $m$. In addition, $r_+(n)$ and $r_-(n)$ denote the transition probabilities that the clone size increases and decreases by one from $n$, respectively, as

$$r_+(n) = \frac{\varepsilon + \lambda n}{\varepsilon K + \lambda N} \times \left(1 - \frac{n}{N}\right), \tag{4}$$

$$r_-(n) = \left(1 - \frac{\varepsilon + \lambda n}{\varepsilon K + \lambda N}\right) \times \frac{n}{N}. \tag{5}$$

The stationary distribution of clone size $P_n^\infty$ was calculated from the following detailed balance:

$$r_-(n)P_n^\infty = r_+(n-1)P_{n-1}^\infty \quad for\ 0 < n \le N, \tag{6}$$

which leads to $P_n^\infty = \{r_+(n-1)/r_-(n)\}P_{n-1}^\infty$. Thus,

$$P_n^\infty = P_0^\infty \prod_{k=1}^{n} \frac{r_+(k-1)}{r_-(k)}. \tag{7}$$

Because $\sum_{n=0}^{N} P_n^\infty = 1$, we obtained

$$P_n^\infty = \prod_{k=1}^{n} \frac{r_+(k-1)}{r_-(k)} \bigg/ \left(1 + \sum_{n=1}^{N} \prod_{k=1}^{n} \frac{r_+(k-1)}{r_-(k)}\right). \tag{8}$$

**First-passage analysis of burst-like clonal expansion**. One of the distinct characteristics of the hNC model was burst-like clonal expansion (Fig. 2c and Supplementary Fig. 1). We analytically evaluated the generation probability and expected duration of a burst. A burst of height $H$ is defined as the dynamics during which the clone size changes from 0 to $H$ without returning to 0, and then returns to 0 without reaching $H + 1$ (Fig. 3b). Briefly, each burst generation can be separated into two processes: the forward process from 0 to $H$ and backward process from $H$ to 0. We evaluated this issue based on the splitting probability and first-passage time.

First, we calculated the probability of generating a burst of height $H$ by multiplying the probabilities of the forward and backward processes. The splitting probability in the forward process can be described as follows:

$$P_{forward}(n;H) = \frac{r_+(n)}{r_+(n)+r_-(n)}P_{forward}(n+1;H) \\ + \frac{r_-(n)}{r_+(n)+r_-(n)}P_{forward}(n-1;H), \tag{9}$$

where $P_{forward}(n;H)$ indicates the splitting probability that the clone size increases from $n$ to $H$ without returning to 0. The boundary conditions are as follows:

$$\begin{cases} P_{forward}(H;H) = 1 \\ P_{forward}(0;H) = 0. \end{cases} \tag{10}$$

$P_{forward}(1;H)$ was analytically solved as

$$P_{forward}(1;H) = \left(1 + \sum_{j=1}^{H-1} \prod_{k=1}^{j} \frac{r_-(k)}{r_+(k)}\right)^{-1}. \tag{11}$$

In the same manner, the splitting probability in the backward process can be described by

$$P_{backward}(n;H) = \frac{r_+(n)}{r_+(n)+r_-(n)}P_{backward}(n+1;H) \\ + \frac{r_-(n)}{r_+(n)+r_-(n)}P_{backward}(n-1;H), \tag{12}$$

where $P_{backward}(n;H)$ indicates the splitting probability of the clone size decreasing from $n$ to 0 without reaching $H$. The boundary conditions are as follows:

$$\begin{cases} P_{backward}(H;H) = 0 \\ P_{backward}(0;H) = 1. \end{cases} \tag{13}$$

$P_{backward}(H;H+1)$ was analytically solved as

$$P_{backward}(H;H+1) = \left(1 + \sum_{j=1}^{H} \prod_{k=j}^{H} \frac{r_+(k)}{r_-(k)}\right)^{-1}. \tag{14}$$

Finally, the generation probability of a burst of height $H$, $P_{gen}(H)$, was obtained using the following equation:

$$P_{gen}(H) = r_+(0) \times P_{forward}(1;H) \times P_{backward}(H;H+1). \tag{15}$$

Second, we calculated the expected duration of a burst of height $H$ by summing the expected durations of the forward and backward processes. The expected duration in the forward process is described by the following recurrence formula:

$$T_{forward}(n;H) = 1 + \left(1 - r_+(n) - r_-(n)\right)T_{forward}(n;H) \\ + r_+(n)T_{forward}(n+1;H) + r_-(n)T_{forward}(n-1;H). \tag{16}$$

where $T_{forward}(n;H)$ indicates the expected duration in which clone size $n$ increases to $H$ without returning to 0 and the first term 1 indicates a unit time increment for a transition. The boundary condition at $n = H$ is $T_{forward}(H;H) = 0$, whereas that at $n = 0$ cannot be defined because of the absorbing boundary; therefore, this

recurrence equation is intractable. To avoid this problem, we introduced the physical quantity $X_{forward}$, defined as

$$X_{forward}(n;H) = P_{forward}(n;H) \times T_{forward}(n;H). \tag{17}$$

Then, the recurrence formula of $X_{forward}(n;H)$ is described by

$$X_{forward}(n;H) = P_{forward}(n;H) + \left(1 - r_+(n) - r_-(n)\right)X_{forward}(n;H) \\ + r_+(n)X_{forward}(n+1;H) + r_-(n)X_{forward}(n-1;H), \tag{18}$$

where the first term represents an increment of $X_{forward}(n;H)$, which occurs with the increment of the unit time. The boundary conditions are:

$$\begin{cases} P_{forward}(H;H) \times T_{forward}(H;H) = 0 \\ P_{forward}(0;H) \times T_{forward}(0;H) = 0. \end{cases} \tag{19}$$

Similarly, we introduced $X_{backward}(n;H) = P_{backward}(n;H) \times T_{backward}(n;H)$ in the backward process, where $T_{backward}(n;H)$ indicates the expected duration in which clone size $n$ decreases to 0 without returning to $H$. The recurrence formula of $X_{backward}(n;H)$ is described by the following equation:

$$X_{backward}(n;H) = P_{backward}(n;H) + \left(1 - r_+(n) - r_-(n)\right)X_{backward}(n;H) \\ + r_+(n)X_{backward}(n+1;H) + r_-(n)X_{backward}(n-1;H), \tag{20}$$

with the following boundary conditions:

$$\begin{cases} P_{backward}(H;H) \times T_{backward}(H;H) = 0 \\ P_{backward}(0;H) \times T_{backward}(0;H) = 0. \end{cases} \tag{21}$$

Finally, the expected duration of a burst of height $H$, $T_{gen}(H)$, was obtained using the following equation:

$$T_{gen}(H) = \frac{1}{r_+(0)} + T_{forward}(1;H) + T_{backward}(H;H+1). \tag{22}$$

**Estimation of parameters in the hNC model from experimental data**. Parameters of the hNC model, i.e., $K$, $N$, and $\varepsilon$, were estimated from the experimental data in a previous study[20]. $K$ was estimated as $K = 420$, which is the number of clone types observed in the experiments. Note that the true value of $K$ might be larger than 420 since the experiment measured only the sampled cells, which are a subset of whole blood cells. For the estimation of $N$ and $\varepsilon$, we proposed two methods, based on the Shannon index and averaged duration of clonal bursts (Fig. 6b, c). $N$ and $\varepsilon$ were estimated based on the Shannon index calculated from the experimental data ($H = 3.64$). In Fig. 6b, when $N$ is larger than 420 ($N > 420$), the Shannon index only depends on $\varepsilon$. Although we cannot exactly determine $N$ from experimental data, we can estimate $\varepsilon \simeq 0.05$ based on the Shannon index. Note that the estimation method based on the average duration of clonal bursts was not applicable to the experimental data of primate hematopoiesis, because the intervals of sampling in the experiment were not small enough to calculate the average duration of clonal bursts.

**Reporting summary**. Further information on research design is available in the Nature Portfolio Reporting Summary linked to this article.

## Data availability
The data used in analysis of primate hematopoiesis is publicly available at Kim et al., *Cell Stem Cell* (2014) (https://doi.org/10.1016/j.stem.2013.12.012). All data generated or analyzed during this study are are available in the github repository, https://github.com/AsahiNakamuta/stemcell.

## Code availability
The mathematical model of clonal expansion (the hierarchical, NC and hNC model) were done using Matlab software (Version R2021b). The Matlab code used for this work can be found at: https://github.com/AsahiNakamuta/stemcell.

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

## Acknowledgements

We are grateful to Dr. Takashi Shinohara and Dr. Mito Kanatsu-Shinohara for their valuable discussions and the editor and reviewers for the informative suggestions related to experimental data, which significantly improved our manuscript. We also thank for the research opportunity provided by the Mathematics-based Creation of Science (MACS) Program in Faculty of Science, Kyoto University (A.N. and H.N.). This study was supported in part by the Moonshot R&D–MILLENNIA Program [grant number JPMJMS2024-9] by JST, Grant-in-Aid for Transformative Research Areas (B) [grant number 21H05170], and Cooperative Study Program of Exploratory Research Center on Life and Living Systems (ExCELLS) [program number 19-102 to H.N.]. It was also supported by JSPS KAKENHI [grant number JP21J23680 to K.Y.] and Grant-in-Aid for Scientific Research (B) [grant number 21H03541 to H.N.], both from the Japan Society for the Promotion of Science (JSPS).

## Author contributions

H.N. conceived the project. A.N., K.Y., and H.N. developed the method, A.N. implemented the model simulation and mathematical analysis, and A.N., K.Y., and H.N. wrote the manuscript.

## Competing interests

The authors declare no competing interests.
