## [Peer Review File · Communications Biology]

Reviewers' comments:

Reviewer #1 (Remarks to the Author):

The manuscript "Stem cell homeostasis regulated by hierarchy and neural competition," by Nakamuta, Yoshido, and Honda present a simple model of stem cell homeostasis and compared it to existing motifs. The authors combine two existing stem cell "motifs", the hierarchical model and the neutral competition model. After analysis and simulations, they describe the behavior of their combined hierarchical neutral competition model and how it scales, relative to the underlying component models. They find that the hierarchical model can exhibit "bursts" and perform simulations and simple analysis via a discrete-time (or generation) master equation. The manuscript is well-written and clear, providing a simple argument for the authors slightly more general model. I recommend publication after the authors consider the following comments and potential improvements:

(1) Clarify the "competition" description of the model earlier by explicitly stating that homeostasis of the TOTAL population is input by hand through a fixed population size, rather than, say, a carrying-capacity in a Lotka-Volterra type model. This will lead to a Moran type model. Also state that N is fixed, etc. Just be a little more explicit in the presentation of the mathematical methods.

(2) Clarify that the "bursts" discussed are really larger scale stochastic fluctuations rather than "dynamics" bursts. Actual, more dynamic bursts can happen as discussed in Xu et al. "Modeling large fluctuations of thousands of clones during hematopoiesis: The role of stem cell self-renewal and bursty progenitor dynamics in rhesus macaque" PLoS Comp. Biol. 14: e1006489, 2018, and other related papers.

In fact, the authors model seems to be the simplest version of progenitor or transit amplifying cells, as discussed in Xu et al. The stemness of stem cells is probably multistage or almost "continuous" and the authors only considered the simplest two-compartment approximation. Is this correct? If so, the authors should mention how multiple stages would make things even more "hierarchical" and mention how their results might qualitatively change.

(3) In the experiments the authors described in their introduction, is sampling an issue? This is probably not an issue for epidermal cells as they are growing in laterally fixed layers, but how would the clone distributions look in small samples of cells?

(4) In their mathematical analysis, the authors considered only the master equation for ONE clone, assuming the others form a background of cells. Is this a mean-field type approximation? Using this

approximation, the authors derived a "stationary distribution" for $m \rightarrow \infty$. However, at very very long times in the Moran model, shouldn't there remain only one clone, with all others going extinct? Can the authors say something about this apparent breakdown of their mean-field result as well as mention this when they present their simulation results?

Reviewer #2 (Remarks to the Author):

In their manuscript entitled "Stem cell homeostasis regulated by hierarchy and neutral competition" Nakamuta and colleagues present a comparative mathematical analysis of two competing stem cell models and a combination thereof. Formulating all models within a consistent mathematical framework the authors present the two "classical" models as limiting cases and study clone size distribution and burst frequencies as characteristic features of all the models. Based on the observed differences the authors suggest strategies to distinguish the models given relevant experimental observations. The simulation results are accompanied by elegant analytical derivations. The paper is (at least in most parts) very well written and the figures are clear and informative. I enjoyed reading it. I wonder a bit about the novelty of the approach and to which extent it really goes beyond what is already known, but in itself it is clear and nice presentation and I do see an added value to the stem cell community. The manuscript would benefit from a stronger coupling to experimental data (see below).

There are a few aspects that the authors should consider when revising their manuscript.

* Coming from the field of hematopoiesis the notion of master stem cells is rather uncommon.

However, there is a hierarchy of long-term and short-term hematopoietic stem cells. There is also a population of multipotent progenitors and it is still not fully clear to which extent those populations can maintain themselves and at the same time contribute to blood production. Also the role of clonal bursts has been discussed before (<https://doi.org/10.1371/journal.pcbi.1006489>). This is a very vivid field also for clonal studies and the authors should at least comment on how their approach maps into this prominent stem cell system.

* While the first pages are really well written and informative, the paragraph on "Experimentally distinguishing the three models" falls behind. It is short and does not meet the level of the previous sections with respect to clarity and understandability. Elaborating a bit more on the two methods and their (experimental) limitations would better round up the manuscript.

I also warrant caution with some of the prerequisites drawn by the authors. Especially the number of introduced label N is usually hardly accessible. It would be very interesting to see how much the estimates of λ depend on this quantity as well as on other model parameters that are not easily measurable.

* The discussion section is in many parts a repetition of aspects mentioned early in the text. I am missing a discussion of how the suggested model integrates with other similar models (e.g. see reference above) and also how it compares to other relevant data sources besides the ones stated already in the introduction/motivation. As for now the model appears somewhat isolated as no quantitative comparison to experimental data is made. This might not be necessary at this point, but a better embedding into the current literature (also with regard to hematopoiesis models) would be helpful.

This is a signed review by Ingmar Glauche

Reviewer #3 (Remarks to the Author):

The manuscript describes a class of models for homeostatic cell fate dynamics which represent a mixture between the classical models of invariant asymmetric divisions (here called hierarchical model) and of population asymmetry (here called neutral competition model), in which stem cells may

divide symmetrically or are lost, yet to equal proportions, so that homeostasis is retained. In their model, there exists a so-called 'master-stem cell' population which divides asymmetrically and produces in a second tier so-called 'neutrally competing' stem cells, which divide and then eventually differentiate. The work analyses this class of models from a generic point to compute expected clone size distributions and clonal burst dynamics. These predictions could then be used in the future, by comparison with the data, to distinguish between different paradigms of homeostatic cell fate choice patterns in tissues.

While the analysis of clonal burst statistics is very promising (in particular with the view to distinguish model paradigms through it), my major concern is that most features of the presented model have already been studied and solved. In particular, the presented model is almost identical (at least in the limit $N \rightarrow \infty$, where $N = \# \text{cells}$ in the open compartment) to the model that has been discussed in [Parigini2020] and the full solution of its Master equation has been determined there (in the Appendix, section "Analysis of the Generalized Invariant Asymmetry model GIA0"). In particular, it has been shown that the clone size distribution can be well approximated by a Gamma distribution, which for small division rate of the 'master stem cell' has a small shape factor and becomes an exponential distribution, exactly as predicted here (which is indeed the same distribution as for neutral competition), while for larger division rates this distribution converges to a Normal distribution. Hence, the aspects related to the general clone size distribution, had already been studied before.

Having said that, the burstiness and the accurate temporal dynamics of clonal bursts has not been studied before, to my knowledge, and hence the authors would be wise to focus on those aspect.

Furthermore, I have a few moderate concerns:

* The definition of 'stem cells' used here is contrary to the most common definition of adult stem cells. In particular, the authors seem to include committed progenitor cells in their definition of stem cells when noting that "...non-master stem cells are more differentiated and irreversibly directed towards differentiation". A commonly accepted defining attribute of adult stem cells is their renewal potential which stands in opposition of being 'irreversibly directed towards differentiation'. Hence, according to common definitions of stem cells, those 'non-master stem cells' cannot be stem cells. Of course, the definitions of stem cells is a disputed field, so I would give the authors all freedom to re-define the concept of stem cells for their purposes, but this should at least be made extremely clear from the outline and should come with a clear benefit compared to the usual stem cell definition.

* The compartment of 'neutrally competing' stem cells in the hNC model cannot be genuinely neutrally competing. If it were, then the additional input of new cells from the 'master stem cell' (M) from the closed compartment, would lead to ever-increasing cell numbers, i.e. would not be homeostatic (see also [Greulich2021]). In fact, in their model, the so-called neutrally competing (NC) stem cells are not neutrally competing at all, since their rate to differentiate, ϵ , is larger than the rate of replacement by another NC cell, which is $\epsilon \cdot N_{NC} / (N_M + N_{NC}) < \epsilon$, where N_{NC} is the number of NC cells and N_M the number of M cells. This means that eventually, any clone starting in the NC compartment will be lost, with exponential rate $e^{-(N_M / (N_M + N_{NC}) \cdot \epsilon \cdot t)}$. This, by the way, also provides a straightforward experimental way to distinguish the hNC from the NC model.

* It is claimed that the hierarchical model is commonly seen as one where the cell type in the 'open compartment' does not divide. This is not congruent with most literature; usually, in the hierarchical model the cell type immediately downstream of the master stem cell is a transit amplifying (TA) cell, i.e. a dividing cell type which is nonetheless committed to eventual differentiation. The latter is similar to the here presented hNC model, although typically, TA cells are seen as losing their proliferative potential in a deterministic sequence, while in the here presented hNC model, this occurs through an intrinsic stochastic bias towards differentiation. However, most would agree that the here presented hNC model would actually fall into the category of hierarchical models, since the base stem cell only undertakes asymmetric divisions.

* The authors claim that they can distinguish which cell fate paradigm (i.e. model) prevails in certain tissue based on cell lineage data. However, they do not demonstrate this. I think it would be not too difficult to test their predictions on existing lineage data from the literature (I don't think additional experiments would be necessary). That would certainly strengthen their claim that their method is working.

In summary, I believe that parts of this manuscript are not novel (the clone size distribution of the model) and I also feel that the authors have not put the model into the correct context, for example due to unfortunate definitions of ('master') 'stem cells' and calling it neutral competition, although it is not genuinely neutral. However, I do see the potential to turn this into a publishable manuscript if the authors focus on the novel aspects of the manuscript, namely the burst dynamics of clones, and how these can be compared with the data in order to distinguish between models. For that however, the authors should also clean up their definitions and put them in context with existing definitions and models of homeostatic tissue self-renewal.

Finally, I have a few minor remarks:

- Grammar: the authors often miss out on articles, "the" and "a". They should do another proof read and possibly get professional help.
- After Eq. (2) on page 7 it doesn't become clear what the index n is.
- Reference 21 doesn't seem to be properly formatted (e.g. journal name missing)

References:

[Parigini2020]: C. Parigini, P. Greulich, Universality of clonal dynamics poses fundamental limits to identify stem cell self-renewal strategies, *eLife* 9:e56532

[Greulich2021]: P. Greulich, B.D. MacArthur, C. Parigini, R. Sanchez-Garcia, Universal principles of lineage architecture and stem cell identity in renewing tissues, *Development* 148:dev194399

Reviewers' comments:

Reviewer #1

The manuscript "Stem cell homeostasis regulated by hierarchy and neural competition," by Nakamuta, Yoshido, and Honda present a simple model of stem cell homeostasis and compared it to existing motifs. The authors combine two existing stem cell "motifs", the hierarchical model and the neutral competition model. After analysis and simulations, they describe the behavior of their combined hierarchical neutral competition model and how it scales, relative to the underlying component models. They find that the hierarchical model can exhibit "bursts" and perform simulations and simple analysis via a discrete-time (or generation) master equation. The manuscript is well-written and clear, providing a simple argument for the authors slightly more general model. I recommend publication after the authors consider the following comments and potential improvements:

We would like to appreciate the reviewer's invaluable comments. We have improved our manuscript and prepared the responses to the reviewer's comments and suggestions as shown below.

(1) Clarify the "competition" description of the model earlier by explicitly stating that homeostasis of the TOTAL population is input by hand through a fixed population size, rather than, asy, a carrying-capacity in a Lotka-Volterra type model. This will lead to a Moran type model. Also state that N is fixed, etc. Just be a little more explicit in the presentation of the mathematical methods.

We apologize for our poor explanation about conditions of the mathematical model. As the reviewer pointed out, the total population of stem cells input by hand in the model simulation, because the loss of one stem cell by differentiation is compensated by supply from another stem cell in the model. We explicitly explained that the total number of competitive stem cells N is fixed and also added the explanation about the similarity and difference with a Lotka-Volterra type model (**Methods, lines 374-377**).

(2) Clarify that the "bursts" discussed are really larger scale stochastic fluctuations rather than "dynamics" bursts. Actual, more dynamic bursts can happen as discussed in Xu et al. "Modeling large fluctuations of thousands of clones during hematopoiesis: The role of stem cell self-renewal and bursty progenitor dynamics in rhesus macaque" PLoS Comp. Biol. 14: e1006489, 2018, and other related papers.

Thank you for telling us an important study, which we did not recognize. We should have mentioned the relationship with this study the reviewer presented, because clonal bursts discussed in our study is large scale stochastic fluctuations rather than dynamical bursts mentioned in that study the reviewer presented. We added the explanation about this difference in **Discussions, lines 266-277**.

In fact, the authors model seems to be the simplest version of progenitor or transit amplifying cells, as discussed in Xu et al. The stemness of stem cells is probably multistage or almost "continuous" and the authors only

considered the simplest two-compartment approximation. Is this correct? If so, the authors should mention how multiple stages would make things even more "hierarchical" and mention how their results might qualitatively change. As mentioned in this comment, it is possible that stem cells consist of multistage and continuous populations, considering that it has recently been revealed that stem cells have heterogeneity among gene expression and the properties as stem cells. We also think that it is necessary to model more complex lineage structure in stem cells in future studies. However, because more complex model requires more assumptions related to lineage structure in detail, which are still controversial in experimental studies, our study assumed only two stem cell populations as a minimal model of stem cell homeostasis in similar with a lot of theoretical studies. In revision, we mentioned this point and the fact that the results we showed possibly change if we assumed that stem cells consist of more multiple differentiation stages (**Discussion, Lines 299-307**).

(3) In the experiments the authors described in their introduction, is sampling an issue? This is probably not an issue for epidermal cells are growing in laterally fixed layers, but how would the clone distributions look in small samples of cells?

We understood that this comment is for experimental systems used in previous studies. Because we did not use experimental data except for data from primate hematopoiesis, we are not sure that we should answer this comment, but we described our opinion below:

A lot of lineage tracing studies of stem cell clones used inducible genetic labeling and transplantation of labeled stem cells, as we mentioned in Introduction (H. J. Snippert et al., 2010, *Cell*; C. Lopez-Garcia et al., 2010, *Science*; E. Clayton et al., 2007, *Nature*; A. M. Klein et al., 2010, *Cell Stem Cell*). In these studies, authors observed labeled stem cells sparsely in tissues (intestinal, epidermal, and spermatogenic stem cells) or in peripheral blood samples (hematopoietic stem cells), because only a limited number of stem cell clones were labeled and sampled from tissues and blood samples. In spite of sparse labeling and observation, it has been regarded that the number of sampling was enough large to represent the actual clone size distribution. Since this point does not seem to affect the results and interpretations in this study, we did not mention in revision. If the reviewer thinks it is necessary to revise our manuscript, we will deal with it shortly.

(4) In their mathematical analysis, the authors considered only the master equation for ONE clone, assuming the others form a background of cells. Is this a mean-field type approximation? Using this approximation, the authors derived a "stationary distribution" for $m \rightarrow \infty$. However, at very very long times in the Moran model, shouldn't there remain only one clone, with all others going extinct? Can the authors say something about this apparent breakdown of their mean-field result as well as mention this when they present their simulation results?

> Is this a mean-field type approximation?

We understand that our master equation approach seems a mean-field approximation, because we focused number of a single clone (n) and treated all other clones as a single variable ($N-n$). But, this is not mean-field approximation. The dynamics of a clone-of-interest can be exactly described by the master equation of number of a single clone, without approximation.

> Using this approximation, the authors derived a "stationary distribution" for $m \rightarrow \infty$. However, at very very long times in the Moran model, shouldn't there remain only one clone, with all others going extinct?

In the NC model, only one clone remains and the others become extinct, as the reviewer mentioned. However, in the hNC model, this situation of domination and extinction did not occur, because master stem cells are without competition and supply competitive stem cells.

Note that domination of single clone observed in the NC model is not phase transition as observed in the Ising spin model, which has been analyzed based on the mean-field approximation.

> Can the authors say something about this apparent breakdown of their mean-field result as well as mention this when they present their simulation results?

As we mentioned in the response to this comment above, we did not use the mean-field approximation and confirmed that analytical solution of the master equation was reproduced by numerical simulation.

Reviewer #2:

In their manuscript entitled “Stem cell homeostasis regulated by hierarchy and neutral competition” Nakamuta and colleagues present a comparative mathematical analysis of two competing stem cell models and a combination thereof. Formulating all models within a consistent mathematical framework the authors present the two “classical” models as limiting cases and study clone size distribution and burst frequencies as characteristic features of all the models. Based on the observed differences the authors suggest strategies to distinguish the models given relevant experimental observations. The simulation results are accompanied by elegant analytical derivations.

The paper is (at least in most parts) very well written and the figures are clear and informative. I enjoyed reading it. I wonder a bit about the novelty of the approach and to which extend it really goes beyond what is already known, but in itself it is clear and nice presentation and I do see an added value to the stem cell community. The manuscript would benefit from a stronger coupling to experimental data (see below).

We would like to thank Dr. Ingmar Glauche for the constructive comments and informative suggestions. We have improved our manuscript and prepared the responses to his comments and suggestions as shown below.

> The manuscript would benefit from a stronger coupling to experimental data (see below).

Thank you for the suggestion. We mentioned this point in the response to his *comment 2*.

There are a few aspects that the authors should consider when revising their manuscript.

Comment 1

* Coming from the field of hematopoiesis the notion of master stem cells is rather uncommon. However, there is a hierarchy of long-term and short-term hematopoietic stem cells. There is also a population of multipotent progenitors and it is still not fully clear to which extend those populations can maintain themselves and at the same time contribute to blood production. Also the role of clonal bursts has been discussed before (<https://doi.org/10.1371/journal.pcbi.1006489>). This is a very vivid field also for clonal studies and the authors should at least comment on **how their approach maps into this prominent stem cell system**.

> Coming from the field of hematopoiesis the notion of master stem cells is rather uncommon. However, there is a hierarchy of long-term and short-term hematopoietic stem cells. There is also a population of multipotent progenitors and it is still not fully clear to which extend those populations can maintain themselves and at the same time contribute to blood production.

Thank you for the important comment about the relationship between stem cell population assumed in our study and that observed in experimental studies. Considering the hierarchical populations within hematopoietic stem cells and their different contribution to the production of differentiated cells (H. Cheng et al., 2020, *Protein Cell*; K. Busch et al., 2015, *Nature*), it is possible that long-term and short-term hematopoietic stem cells correspond to master and competitive stem cells in the hNC model, respectively. In revision, we mentioned this possibility in **Discussion Lines**

260-265. We also referred to the possibility of more multistage stem cell populations in hematopoiesis in the response to *Comment 2 by Reviewer #1*.

> Also the role of clonal bursts has been discussed before (<https://doi.org/10.1371/journal.pcbi.1006489>). This is a very vivid field also for clonal studies and the authors should at least comment on how their approach maps into this prominent stem cell system.

Thank you for telling us an important study, which we did not recognize. In revision, we compared our results with data from the experimental study of the primate hematopoiesis and showed that our mathematical model generated similar clonal bursts and clone size distribution with the experimental results in primate hematopoiesis (**Results, Fig. 5, Lines 197-213**). We also mentioned that it is possible that clonal bursts could be generated by different mechanism suggested in previous study, which the reviewer mentioned in this comment (**Discussion, Lines 266-277**).

Comment 2

* While the first pages are really well written and informative, the paragraph on “Experimentally distinguishing the three models” falls behind. It is short and does not meet the level of the previous sections with respect to clarity and understandability. Elaborating a bit more on the two methods and their (experimental) limitations would better round up the manuscript.

We apologize for the poor description in that paragraph. In revision, we newly demonstrated to estimate the proliferation rate of master stem cells ϵ from the experimental data of primate hematopoiesis (S. Kim et al., 2014, *Cell Stem Cell*), which Dr. Glauche (and the reviewer #1) suggested (**Results, Lines 241-245**). In addition, we analyzed the indexes for the estimation of ϵ (Shannon index and average duration of all bursts) in more detail as the reviewer suggested (see below).

I also warrant caution with some of the prerequisites drawn by the authors. **Especially the number if introduced label N is usually hardly accessible.** It would be very interesting to see how much the estimates of ϵ depend on this quantity as well as on other model parameters that are not easily measurable.

> It would be very interesting to see how much the estimates of ϵ depend on this quantity as well as on other model parameters that are not easily measurable.

Thank you for the suggestion. As suggested, we examined the dependency of two indexes (Shannon index and average duration of all bursts) on the proliferation rate of master stem cells ϵ and the total number of competitive stem cells N (Fig. 6, **Results, Lines 234-240**).

> I also warrant caution with some of the prerequisites drawn by the authors. **Especially the number if introduced label N is usually hardly accessible.**

We also thank the reviewer for mentioning the important point that N is usually hard to measure from experiments. Theoretically, if we measure both the Shannon index and the average duration of all clonal bursts, we

can estimate both ε and N . However, it was difficult to measure the average duration of all clonal bursts from experimental data in primate hematopoiesis because the intervals of sampling in the experiment were not small enough to calculate the average duration of clonal bursts. In revision, we showed that it is possible to estimate ε , without knowing the exact number of N (**Results, Lines 241-245, and Methods, Lines 448-457**).

Comment 3

* The discussion section is in many parts a repetition of aspects mentioned early in the text. I am missing a discussion of how the suggested model integrates with other similar models (e.g. see reference above) and also how it compares to other relevant data sources besides the ones stated already in the introduction/motivation. As for now the model appears somewhat isolated as no quantitative comparison to experimental data is made. This might not be necessary at this point, but a better embedding into the current literature (also with regard to hematopoiesis models) would be helpful.

We apologize for the poor discussion about the comparison with previous studies. In revision, we deleted the redundant sentences and modified the discussion, and integrated two paragraphs (“*Reconsideration of the hierarchical and NC model*” and “*Comparison between the NC and hNC models*”) into one paragraph (“*Reconsideration of the hierarchical and NC model*”). In addition, we mentioned the relationship between our results with the previous experimental and theoretical studies below:

1. M. Klose, M. C. Florian, A. Gerbaulet, H. Geiger, I. Glauche, Hematopoietic Stem Cell Dynamics Are Regulated by Progenitor Demand: Lessons from a Quantitative Modeling Approach. *Stem Cells* **37**, 948-957 (2019)
2. S. Xu, S. Kim, I. S. Y. Chen, T. Chou, Modeling large fluctuations of thousands of clones during hematopoiesis: The role of stem cell self-renewal and bursty progenitor dynamics in rhesus macaque. *PLoS Comput Biol* **14**, e1006489 (2018)
3. C. Parigini, P. Greulich, Universality of clonal dynamics poses fundamental limits to identify stem cell self-renewal strategies. *Elife* **9**, (2020)
4. P. Greulich, B. D. MacArthur, C. Parigini, R. J. Sánchez-García, Universal principles of lineage architecture and stem cell identity in renewing tissues. *Development* **148**, (2021).

Regarding the former two studies, in revision, we showed the consistency with the experimental results in primate hematopoiesis and the difference of the mechanism. We mentioned it in detail in response to **Comment 1**. Regarding the latter two studies, in revision, we mentioned that our mathematical model was consistent with that proposed in these studies because they showed that clone size distribution in the model with hierarchy among stem cells also followed scaling law depending on the model parameters (**Discussion, Lines 327-336**). We referred to this point in detail in response to **Comment 1 by Reviewer #3**.

> As for now the model appears somewhat isolated as no quantitative comparison to experimental data is made. This might not be necessary at this point, but a better embedding into the current literature (also with regard to hematopoiesis models) would be helpful.

In revision, we compared the results from our simulations with experimental data in primate hematopoiesis. Although

it was difficult to compare quantitatively, we showed the consistency in terms of the existence of clonal bursts and clone size distribution (**Results, Lines 197-213**).

Reviewer #3:

The manuscript describes a class of models for homeostatic cell fate dynamics which represent a mixture between the classical models of invariant asymmetric divisions (here called hierarchical model) and of population asymmetry (here called neutral competition model), in which stem cells may divide symmetrically or are lost, yet to equal proportions, so that homeostasis is retained. In their model, there exists a so-called 'master-stem cell' population which divides asymmetrically and produces in a second tier so-called 'neutrally competing' stem cells, which divide and then eventually differentiate. The work analyses this class of models from a generic point to compute expected clone size distributions and clonal burst dynamics. These predictions could then be used in the future, by comparison with the data, to distinguish between different paradigms of homeostatic cell fate choice patterns in tissues.

We would like to appreciate the reviewer's invaluable comments. We have improved our manuscript and prepared the responses to the reviewer's comments and suggestions as shown below.

Comment 1

While the analysis of clonal burst statistics is very promising (in particular with the view to distinguish model paradigms through it), my major concern is that most features of the presented model have already been studied and solved. In particular, the presented model is almost identical (at least in the limit $N \rightarrow \infty$, where $N = \#$ cells in the open compartment) to the model that has been discussed in [Parigini2020] and the full solution of its Master equation has been determined there (in the Appendix, section "Analysis of the Generalized Invariant Asymmetry model GIA0"). In particular, it has been shown that the clone size distribution can be well approximated by an Gamma distribution, which for small division rate of the 'master stem cell' has a small shape factor and becomes an exponential distribution, exactly as predicted here (which is indeed the same distribution as for neutral competition), while for larger division rates this distribution converges to a Normal distribution. Hence, the aspects related to the general clone size distribution, had already been studied before.

We apologize for not having recognized the previous study, which proposed the general model of stem cell homeostasis. Indeed, Parigini and Greulich derived the clone size distribution from the master equation and it followed scaling law even if lineage structure has hierarchy among stem cells (the generalized invariant asymmetry model; GIA0 model), depending on model parameters. However, the parameter of the GIA0 model showing scaling law in clone size distribution (C. Parigini, P. Greulich, 2020, *eLife*) is different from the condition in the hNC model; proliferation rates of renewing cells (i.e., master stem cells) and committed cells (i.e., competitive stem cells) are the same in the GIA0 model, whereas proliferation rate of master stem cells is greatly smaller than that of competitive stem cells in the hNC model. In revision, we mentioned that the findings in the previous studies (C. Parigini, P. Greulich, 2020, *eLife*; P. Greulich et al., 2021, *Development*) is consistent with our observation of scaling law of clone size distribution in the hNC model in that scaling law cannot rule out the possibility of hierarchy among stem cells (**Discussion, Lines 327-336**).

Having said that, the burstiness and the accurate temporal dynamics of clonal bursts has not been studied before, to my knowledge, and hence the authors would be wise to focus on those aspect.

Thank you for your appreciation. In revision, we further analyzed the experimental data (S. Kim et al., 2014, *Cell Stem Cell*) by focusing on the burstiness (Fig. 5A and B).

Furthermore, I have a few moderate concerns:

Comment 2

The definition of 'stem cells' used here is contrary to the most common definition of adult stem cells. In particular, the authors seem to include committed progenitor cells in their definition of stem cells when noting that "...non-master stem cells are more differentiated and irreversibly directed towards differentiation". A commonly accepted defining attribute of adult stem cells is their renewal potential which stands in opposition of being 'irreversibly directed towards differentiation'. Hence, according to common definitions of stem cells, those 'non-master stem cells' cannot be stem cells. Of course, the definitions of stem cells is a disputed field, so I would give the authors all freedom to re-define the concept of stem cells for their purposes, but this should at least be made extremely clear from the outline and should come with a clear benefit compared to the usual stem cell definition.

Thank you for mentioning the important point of the difficulty in the definition of stem cells. We understand that the cells which irreversibly directed towards differentiation cannot usually be classified into stem cells. However, it has been recently considered that stem cells have heterogeneity among gene expression and the properties as stem cells (H. Cheng et al., 2020, *Protein Cell*; R. Jurecic, 2019, *Adv Exp Med Biol*). Thus, we considered that it is more valid that master stem cells supply other types of stem cells, even if stem cell homeostasis is regulated by the hierarchical model. Notably, it is similar definition of stem cells used in previous studies of intestinal stem cells (H. J. Snippert et al., 2010, *Cell*; C. Lopez-Garcia et al., 2010, *Science*). In revision, we mentioned this point in **Discussion, Lines 298-317**.

Comment 3

The compartment of 'neutrally competing' stem cells in the hNC model cannot be genuinely neutrally competing. If it were, then the additional input of new cells from the 'master stem cell' (M) from the closed compartment, would lead to ever-increasing cell numbers, i.e. would not be homeostatic (see also [Greulich2021]). In fact, in their model, the so-called neutrally competing (NC) stem cells are not neutrally competing at all, since their rate to differentiate, ϵ , is larger than the rate of replacement by another NC cell, which is $\epsilon \cdot N_{NC} / (N_M + N_{NC}) < \epsilon$, where N_{NC} is the number of NC cells and N_M the number of M cells. This means that eventually, any clone starting in the NC compartment will be lost, with exponential rate $e^{-(N_M / (N_M + N_{NC}) \cdot \epsilon \cdot t)}$. This, by the way, also provides a straightforward experimental way to distinguish the hNC from the NC model.

Due to our poor explanation, the reviewer seems to misunderstand that loss of competitive stem cells is compensated mainly by master stem cells. We assumed that proliferation of master stem cells λ is smaller than that of competitive stem cells λ in the hNC model, which could be the biologically valid condition. In the hNC model, the rate of

replacement by competitive stem cells (called NC cells in this comment) is larger than that of replacement by master stem cell (called M cells in this comment);

- the rate of replacement by another NC cell $= \lambda * N_{NC} / (\epsilon * N_M + \lambda * N_{NC})$
- the rate of replacement by master stem cell $= \epsilon * N_M / (\epsilon * N_M + \lambda * N_{NC})$

Therefore, competitive stems are mainly neutrally competing. In revision, we explicitly mentioned that the condition $\lambda > \epsilon$ is biologically reasonable in the hNC model (**Results, Lines 110-112**).

Comment 4

* It is claimed that the hierarchical model is commonly seen as one where the cell type in the 'open compartment' does not divide. This is not congruent with most literature; usually, in the hierarchical model the cell type immediately downstream of the master stem cell is a transit amplifying (TA) cell, i.e. a dividing cell type which is nonetheless committed to eventual differentiation. The latter is similar to the here presented hNC model, although typically, TA cells are seen as losing their proliferative potential in a deterministic sequence, while in the here presented hNC model, this occurs through an intrinsic stochastic bias towards differentiation. However, most would agree that the here presented hNC model would actually fall into the category of hierarchical models, since the base stem cell only undertakes asymmetric divisions.

We understand that downstream of master stem cells is generally considered to be a TA cell. As we mentioned in response to **Comment 2**, however, we considered that it is more valid that master stem cells supply other types of stem cells even in the hierarchical model due to the heterogeneity among stem cells. In addition, in the framework of generalized model suggested in previous studies (C. Parigini, P. Greulich, 2020, *eLife*; P. Greulich et al., 2021, *Development*), the hNC model can be included in the hierarchical model. However, we classified the model (the hierarchical, NC, and hNC model) simply based on the presence of master stem cells, which undergo invariant asymmetric divisions, and neutral competition among stem cells. We referred to this point in **Discussion, Lines 308-336**.

Comment 5

The authors claim that they can distinguish which cell fate paradigm (i.e. model) prevails in certain tissue based on cell lineage data. However, they do not demonstrate this. I think it would be not too difficult to test their predictions on existing lineage data from the literature (I don't think additional experiments would be necessary). That would certainly strengthen their claim that their method is working.

Thank you for the important suggestion. In revision, we analyzed data from primate hematopoiesis published by Kim et al (S. Kim et al., 2014, *Cell Stem Cell*). We mentioned that hematopoietic stem cells in primate was possibly regulated by the hNC model, because bursts were observed in long-term clonal tracking experiment. In addition, we estimated the proliferation of master stem cells $\epsilon \approx 0.05$ from the data, which supported the possibility of the hNC model. We mentioned the comparison between our model and experimental data in primate hematopoiesis in **Results, Lines 197-213**, and the estimation of ϵ from the data in **Results, Lines 241-245** and **Methods, Lines 448-457**.

Summary

In summary, I believe that parts of this manuscript are **not novel** (the clone size distribution of the model), and I also feel that the authors have not put the model into the correct context, for example due to unfortunate definitions of ('master') 'stem cells" and calling it neutral competition, although it is not genuinely neutral. However, I do see the potential to turn this into a publishable manuscript if the authors focus on the novel aspects of the manuscript, namely the burst dynamics of clones, and how these can be compared with the data in order to distinguish between models. For that however, the authors should also clean up their definitions and put them in context with existing definitions and models of homeostatic tissue self-renewal.

> parts of this manuscript are **not novel** (the clone size distribution of the model)

We addressed this point in response to Comment 1.

> due to unfortunate definitions of ('master') 'stem cells" and calling it neutral competition, although it is not genuinely neutral.

We addressed this point in response to Comment 2 and 3.

> I do see the potential to turn this into a publishable manuscript if the authors focus on the novel aspects of the manuscript, namely the burst dynamics of clones,

We addressed this point in response to Comments 1.

> how these can be compared with the data in order to distinguish between models.

We addressed this point in response to Comments 5.

Minor remarks:

- Grammar: the authors often miss out on articles, "the" and "a". They should do another proof read and possible get professional help.

Sorry for mistakes in grammar. After revision, we used a professional English proofreading.

- After Eq. (2) on page 7 it doesn't become clear what the index n is.

Sorry for using confusing notations. In revision, we changed the index in Eq. (2) for readers to understand easily.

- Reference 21 doesn't seem to be properly formatted (e.g. journal name missing)

We apologize for the mistake in Reference. We revised it in revision.

Reviewers' comments:

Reviewer #1 (Remarks to the Author):

This revised manuscript is a significant improvement over the first version, and it includes additional simulations and comparisons with data. I believe this version can be published as it provides a much fuller context of their modeling. I would request that the authors consider:

The comparison between Fig 5(a) and (b) do not seem very good. Can the authors describe how they chose the parameters? Was there some fitting? If so, what was the fitting procedure? If parameters were just chosen to provide a qualitative comparison, that should be stated and some discussion of how changing certain parameters would affect the simulation results discussed.

Since the authors compared to rhesus macaque data, they should clarify the fact that the data arises from a small sample of blood from the animals, whereas their model seems to refer to the whole organism? The sampling is also discussed in Song et al.

If the authors put some thought into the above remaining questions and address them before they submit their final files, I can recommend acceptance without having to review it again.

Reviewer #2 (Remarks to the Author):

I was pleased to reassess the revised version of the manuscript entitled "Stem cell homeostasis regulated by hierarchy and neutral competition" by Nakamuta and colleagues. The authors did a good job to better put their work in the scientific context, especially with regard to hematopoiesis. Their analysis of hematopoietic burst in a respective primate data set is a real plus and I appreciate the additional results and figures. The respective sections could have been integrated a bit smoother in the overall outline, but it is fair enough to see how this part entered during the revision process. I agree with reviewer 3 that the naming convention for the cell types is challenging. Non-master stem cells could also be referred to as progenitors. However, I feel that the examples explain the general notion and the mapping should be left to meet the conventions of a particular stem cell system. As a final minor comment: Rather than only providing a github link, the authors might share which software they used for their implementations.

Reviewer #3 (Remarks to the Author):

The revised manuscript and the authors' rebuttal address my concerns, and the new manuscript is certainly an improvement: the manuscript is now set better within the literature context and the their fleshed-out study how to distinguish the models experimentally, in particular by giving an explicit example with data from primates, provides added value.

While the manuscript has improved, there are still some unsatisfactory parts and incorrect statements in the manuscript and rebuttal.

First and foremost, the authors claim in their rebuttal that the solution of the Master equation done in Ref (Parigini+Greulich 2020) only considers a scaling limit where the cell division rate of the master stem cell (ϵ) is of equal magnitude as that of the second-tier proliferative cells (λ). This is not true. The treatment covers general cell division rates and considers several limits, including that where ϵ is much lower than that λ , which corresponds to the hNC model. Hence, this case has been studied, and also, an exponential clone size distribution has been predicted already. What is

true is that this case is only considered as a side remark in the main text (Parigini+Greulich 2020) and is not classified as a separate model class. The point is that for any fixed epsilon, lambda, the here-presented hNC model corresponds to one from the GIA class as introduced in (Parigini+Greulich 2020).

Having said that, I do think that the authors have a point in considering the special case $\epsilon/\lambda \rightarrow 0$ while $\epsilon > 0$, as its behaviour is sufficiently different to the case general hierarchical case. So it could make sense to speak of a third class of models; however, the differences are quite subtle here (it is about the difference between a small value vs. a genuine limit to zero), and I feel that currently the discussion in the presented manuscript is too vague in that aspect. The authors do explain this to a reasonable accurate degree in their rebuttal, but I miss some rigour in the manuscript.

Second, I still find it misleading to speak of the second-tier proliferative cells as stem cells. After all, their total differentiation rate ($\lambda + \epsilon/r$, where r is the ratio of second-tier proliferative cells over master stem cells) is slightly higher than their proliferation rate (λ). If one would label the second-tier proliferative cells by a genetic marker, their progeny would -- possibly very slowly but steadily -- be declining and eventually be replaced by the progeny of the master stem cell. This means that second-tier proliferative cells are not self-renewing (yes, they *would be* self-renewing if there weren't any master stem cells, but the matter of stemness must be seen in the context of the cellular environment). Hence, they are "almost renewing" but not entirely. Furthermore, those second-tier proliferative cells don't have full lineage potential, as they do not generate master stem cells. Thus they are lacking the two fundamental characteristics of stem cells, namely self-renewal potential and full lineage potential.

Again, I do see some merit in the authors' discussion of this, since the matter of being "almost" self-renewing, might not be experimentally distinguishable in the case that the division rate of the master stem cell is sufficiently low, and the ratio of second-tier proliferative cells over master stem cells is sufficiently high. However, again, I do miss a rigorous discussion of these subtle issues; the discussion remains vague.

Hence, in my view the manuscript contains valuable insights, but these are not sufficiently carved out of the context of other studies (and with respect to the common definition of adult stem cells) which studied very similar models already. The differences are subtle but relevant, and need to be further worked out, since the hNC model and its full solution are not novel on their own.

Reviewers' comments:

Reviewer #1

This revised manuscript is a significant improvement over the first version, and it includes additional simulations and comparisons with data. I believe this version can be published as it provides a much fuller context of their modeling. I would request that the authors consider:

We would like to appreciate the reviewer's invaluable comments. We have improved our manuscript and prepared the responses to the reviewer's comments and suggestions as shown below.

Comment 1

The comparison between Fig 5(a) and (b) do not seem very good. Can the authors describe how they chose the parameters? Was there some fitting? If so, what was the fitting procedure? If parameters were just chosen to provide a qualitative comparison, that should be stated and some discussion of how changing certain parameters would affect the simulation results discussed.

As we mentioned in Methods (Methods, lines 446-456), we estimated ϵ and N . The estimation was based on the Shannon index, not by a qualitative comparison. In revision, we mentioned this point more clearly (Methods, line 454).

Comment 2

Since the authors compared to rhesus macaque data, they should clarify the fact that the data arises from a small sample of blood from the animals, whereas their model seems to refer to the whole organism? The sampling is also discussed in Song et al.

Thank you for your valuable comment. In revision, we clearly mentioned the fact that data of primate hematopoiesis was based on sampling of only a part of blood cells (Methods, lines 448-450)

If the authors put some thought into the above remaining questions and address them before they submit their final files, I can recommend acceptance without having to review it again.

Thank you for your valuable comments and suggestions that have greatly helped us improve our manuscript.

Reviewer #2:

I was pleased to reassess the revised version of the manuscript entitled “Stem cell homeostasis regulated by hierarchy and neutral competition” by Nakamuta and colleagues. The authors did a good job to better put their work in the scientific context, especially with regard to hematopoiesis. Their analysis of hematopoietic burst in a respective primate data set is a real plus and I appreciate the additional results and figures. The respective sections could have been integrated a bit smoother in the overall outline, but it is fair enough to see how this part entered during the revision process.

I agree with reviewer 3 that the naming convention for the cell types is challenging. Non-master stem cells could also be referred to as progenitors. However, I feel that the examples explain the general notion and the mapping should be left to meet the conventions of a particular stem cell system. As a final minor comment: Rather than only providing a github link, the authors might share which software they used for their implementations.

We would like to appreciate the reviewer’s invaluable comments. In revision, we mentioned the software used in addition to the GitHub link (Code availability, lines 463-465).

Reviewer #3:

The revised manuscript and the authors' rebuttal address my concerns, and the new manuscript is certainly an improvement: the manuscript is now set better within the literature context and the their fleshed-out study how to distinguish the models experimentally, in particular by giving an explicit example with data from primates, provides added value.

We would like to appreciate the reviewer’s invaluable comments. We have improved our manuscript and prepared the responses to the reviewer’s comments and suggestions as shown below.

While the manuscript has improved, there are still some unsatisfactory parts and incorrect statements in the manuscript and rebuttal.

Comment 1

First and foremost, the authors claim in their rebuttal that the solution of the Master equation done in Ref (Parigini+Greulich 2020) only considers a scaling limit where the cell division rate of the master stem cell (ϵ) is of equal magnitude as that of the second-tier proliferative cells (λ). This is not true. The treatment covers general cell division rates and considers several limits, including that

where epsilon is much lower than that lambda, which corresponds to the hNC model. Hence, this case has been studied, and also, an exponential clone size distribution has been predicted already. What is true is that this case is only considered as a side remark in the main text (Parigini+Greulich 2020) and is not classified as a separate model class. The point is that for any fixed epsilon, lambda, the here-presented hNC model corresponds to one from the **GIA** class as introduced in (Parigini+Greulich 2020).

As the reviewer pointed out, our model is indeed included in the GIA model. We have already mentioned this point (Discussion, lines 326-336), and added the explicit explanation in revision (Discussion, lines 326-327). We consider the novelty of our study as the point that we showed that clonal bursts were generated by the hNC model and proposed the experimental way of distinguishing models based on the distinct behaviors.

Having said that, I do think that the authors have a point in considering the special case epsilon/lambda $\rightarrow 0$ while epsilon > 0 , as its behaviour is sufficiently different to the case general hierarchical case. So it could make sense to speak of a third class of models; however, the differences are quite subtle here (it is about the difference between a small value vs. a genuine limit to zero), and I feel that currently the discussion in the presented manuscript is too vague in that aspect. The authors do explain this to a reasonable accurate degree in their rebuttal, but I miss some rigour in the manuscript.

Thank you for mentioning the important point. In revision, we clearly mentioned that we considered the NC and hNC model as different biological models, because these two models showed the distinct behaviors in clonal expansion, as mentioned in our previous rebuttal (Discussion, lines 252-256).

Comment 2

Second, I still find it misleading to speak of the second-tier proliferative cells as stem cells. After all, their total differentiation rate ($\lambda + \epsilon/r$, where r is the ratio of second-tier proliferative cells over master stem cells) is slightly higher than their proliferation rate (λ). If one would label the second-tier proliferative cells by a genetic marker, their progeny would -- possibly very slowly but steadily -- be declining and eventually be replaced by the progeny of the master stem cell. This means that second-tier proliferative cells are not self-renewing (yes, they *would be* self-renewing if there weren't any master stem cells, but the matter of stemness must be seen in the context of the cellular environment). Hence, they are "almost renewing" but not entirely. Furthermore, those second-tier proliferative cells don't have full lineage potential, as they do not generate master stem cells. Thus they are lacking the two fundamental characteristics of stem cells, namely self-renewal potential and

full lineage potential.

Again, I do see some merit in the authors' discussion of this, since the matter of being "almost" self-renewing, might not be experimentally distinguishable in the case that the division rate of the master stem cell is sufficiently low, and the ratio of second-tier proliferative cells over master stem cells is sufficiently high. However, again, I do miss a rigorous discussion of these subtle issues; the discussion remains vague.

We understand that competitive stem cells in the hNC model (second-tier proliferative stem cells) might not be named 'stem cells' because they lack self-renewal potential and full lineage potential. Still, as we mentioned about the reason why we regarded non-master stem cells as stem cells in the hierarchical model, it is more feasible that master stem cells supply other types of stem cells, considering more heterogeneous nature of stem cell populations than previously expected. Therefore, we assumed two compartments of stem cells in the hNC model, which is based on the similar idea in previous studies of intestinal stem cells (H. J. Snippert et al., 2010, *Cell*; C. Lopez-Garcia et al., 2010, *Science*) and consistent with the suggestion by the reviewer #2 that we should follow the conventions of a particular stem cell system. We mentioned our definition of stem cells in the section of '*Biological relevance of our mathematical model*' in Discussion (Discussion, lines 298-318).

Hence, in my view the manuscript contains valuable insights, but these are not sufficiently carved out of the context of other studies (and with respect to the common definition of adult stem cells) which studied very similar models already. The differences are subtle but relevant, and need to be further worked out, since the hNC model and its full solution are not novel on their own.

Thank you for your valuable comments. We understand that our model is similar to GIA model, but we showed clonal burst dynamics, which has never been reported by previous theoretical models. Thus, we believe that our finding shed light on our biological understanding of stem cell homeostasis.